

# How has our knowledge of dinosaur diversity through geologic time changed through research history?

Jonathan P. Tennant[1], Alfio Alessandro Chiarenza[1] and Matthew Baron[2,3]

[1] Department of Earth Science and Engineering, Imperial College London, London, UK
[2] Department of Earth Science, University of Cambridge, Cambridge, UK
[3] Earth Sciences Department, Natural History Museum, London, UK

## ABSTRACT

Assessments of dinosaur macroevolution at any given time can be biased by the historical publication record. Recent studies have analysed patterns in dinosaur diversity that are based on secular variations in the numbers of published taxa. Many of these have employed a range of approaches that account for changes in the shape of the taxonomic abundance curve, which are largely dependent on databases compiled from the primary published literature. However, how these 'corrected' diversity patterns are influenced by the history of publication remains largely unknown. Here, we investigate the influence of publication history between 1991 and 2015 on our understanding of dinosaur evolution using raw diversity estimates and shareholder quorum subsampling for the three major subgroups: Ornithischia, Sauropodomorpha, and Theropoda. We find that, while sampling generally improves through time, there remain periods and regions in dinosaur evolutionary history where diversity estimates are highly volatile (e.g. the latest Jurassic of Europe, the mid-Cretaceous of North America, and the Late Cretaceous of South America). Our results show that historical changes in database compilation can often substantially influence our interpretations of dinosaur diversity. 'Global' estimates of diversity based on the fossil record are often also based on incomplete, and distinct regional signals, each subject to their own sampling history. Changes in the record of taxon abundance distribution, either through discovery of new taxa or addition of existing taxa to improve sampling evenness, are important in improving the reliability of our interpretations of dinosaur diversity. Furthermore, the number of occurrences and newly identified dinosaurs is still rapidly increasing through time, suggesting that it is entirely possible for much of what we know about dinosaurs at the present to change within the next 20 years.

## INTRODUCTION

In the latter half of the 20th century, palaeobiology underwent a renaissance by adopting a more quantitative analytical approach to understanding changes in the fossil record through time (*Valentine & Moores, 1970*; *Raup, 1972*; *Gould & Eldredge, 1977*;

Corresponding authors
Jonathan P. Tennant,
jon.tennant.2@gmail.com
Alfio Alessandro Chiarenza,
a.chiarenza15@imperial.ac.uk

*Sepkoski et al., 1981*; *Van Valen, 1984*; *Sepkoski, 1996*). This research was largely focussed around estimating patterns of animal diversity, extinction and speciation through time, and what the external processes governing these were. To this day, reconstructing the diversity of life through geological time remains one of the most crucial aspects of palaeobiology, as it allows us to address broader questions about the evolution of life and what the mechanisms of extinction and recovery are. These pioneering analyses were largely based on an archive of range-through taxa of marine animals, known as the 'Sepkoski Compendium'. More recently, analytical palaeobiology has had a second wave of innovation, in part due to development of large databases that catalogue fossil occurrences and associated data such as the Paleobiology Database (www.paleobiodb.org), and also due to development of increasingly sophisticated analytical subsampling (*Alroy, 2000a*, *2003*, *2010a*; *Starrfelt & Liow, 2016*) and modelling (*Smith & McGowan, 2007*; *Lloyd, 2012*) techniques. Together, these are helping to provide new insight into how we can use the fossil record to understand the large-scale evolutionary patterns and processes that have shaped the history of life.

All of these studies, both older and more recent, are underpinned by a single principle, in that they rely on the recorded number of identifiable fossiliferous occurrences present through geological time. Despite meticulous work to ensure that these databases and compendia represent the best possible records of historical trends, there has been continuous discussion as to the accuracy of the data, and the extent to which estimates of palaeodiversity might be confounded by such bias. These biases include factors such as heterogeneous sampling intensity, fossiliferous rock availability, and variable depth of taxonomic research (*Raup, 1972*, *1976*; *Uhen & Pyenson, 2007*; *Benton, 2008a*, *2008b*; *Marx & Uhen, 2010*; *Tarver, Donoghue & Benton, 2011*; *Smith, Lloyd & McGowan, 2012*; *Smith & Benson, 2013*).

In 1993, Sepkoski added an additional dimension to these studies by assessing how database compilation history through changes in taxonomy, stratigraphic resolution, and sampling influences the shape of macroevolutionary patterns (*Sepkoski, 1993*). Based on comparison of the two compendia built in 1982 and 1992, *Sepkoski (1993)* found that in spite of numerous taxonomic changes over 10 years, the overall patterns of diversity for marine animals remained relatively constant, with the main notable change being that overall diversity was consistently higher in the 1992 compilation. *Alroy (2000a)* similarly showed that database age does appear to have an influence on North American mammal diversity estimates, and *Alroy (2010c)* further demonstrated that diversity estimates based on data from the Paleobiology Database were proportionally similar to either the genus- or family-level results based on Sepkoski's original compendium. At the present, there are three main arguments regarding the historical reliability of diversity curves (*Sepkoski et al., 1981*; *Sepkoski, 1993*; *Alroy, 2000a*): firstly that because independent datasets produce similar diversity curves, this suggests that convergence on a common signal reflecting either a real evolutionary, fossil record structure, or taxonomic phenomenon; second, that the addition of new data to existing compilations should yield only minor changes to resulting diversity estimates; and third, that the addition of new data can potentially dramatically alter the shape of diversity (counter to the first and

second arguments). At the present, the first argument appears to be the best supported by analytical evidence (*Sepkoski, 1993*; *Alroy, 2000b*).

However, besides Sepkoski and Alroy's work, relatively little consideration has been given to how publication or database history can influence macroevolutionary patterns, despite an enormous reliance on their research utility (although see *Benton (2008a*, *2008b)* and *Tarver, Donoghue & Benton (2011)* for examples using vertebrates). In particular, to our knowledge, no one has yet tested this potential influence using an occurrence-based tetrapod dataset, such as those available from the Paleobiology Database. This is important, given that a wealth of recent studies, and in particular on tetrapod groups, have focussed on estimating diversity patterns through geological time and interpreting what the potential drivers of these large-scale evolutionary patterns might be (*Butler et al., 2009*; *Benson & Butler, 2011*; *Butler et al., 2011*; *Mannion et al., 2015*; *Nicholson et al., 2015*; *Benson et al., 2016*; *Grossnickle & Newham, 2016*; *Nicholson et al., 2016*; *Tennant, Mannion & Upchurch, 2016a*; *Brocklehurst et al., 2017*). Many of these studies have employed subsampling methods that are sensitive to changes in the shape of the taxonomic abundance distribution, which we would expect to change in a non-random fashion based on new fossil discoveries through time as they are published (*Benton et al., 2011*, *2013*; *Benton, 2015*) (e.g. due to the opening up of new discovery regions for geopolitical reasons, or the historical and macrostratigraphic availability of fossil-bearing rock formations). Furthermore, as sampling increases through time, we might also expect the relative proportion of singleton occurrences to decrease, improving the evenness of the underlying sampling pool (*Alroy, 2010a*; *Chao & Jost, 2012*), and therefore influencing calculated diversity estimates (see 'Methods' below). Assessing this influence in a historical context is therefore important for understanding how stable our interpretations of evolutionary patterns are.

While the data used in these analyses are typically based on a 'mature' dataset that has undergone rigorous taxonomic scrutiny and data addition or refinement, they often tend to neglect explicit consideration of the potential influence of temporal variations in the publication record (which these databases are explicitly based on). This has important implications for several reasons. First we might expect the shape of both raw and subsampled diversity curves to change through time in concert with new discoveries and as sampling increases (*Sepkoski, 1993*; *Alroy, 2000a*), or that subsampled diversity estimates stabilise at some point. Second, this could therefore impact our interpretations of the relative magnitude, tempo and mode of apparent radiations and extinctions. Third, if the shape of estimated diversity curves change (either based on raw or 'corrected' data), we could see that the strength of results from comparisons of diversity with extrinsic factors such as sea-level or palaeotemperature (*Benson et al., 2010*; *Benson & Butler, 2011*; *Butler et al., 2011*; *Peters & Heim, 2011b*; *Mayhew et al., 2012*; *Martin et al., 2014*; *Mannion et al., 2015*; *Nicholson et al., 2015*; *Tennant, Mannion & Upchurch, 2016a*, *2016b*) will change.

As our data become updated, capturing this influence of sampling variation becomes more important through longer periods of time. We might expect sampling error to be highest earlier on in sampling history, and to reduce through time, therefore improving

the reliability of our correlation estimates. However, if our subsampled diversity estimates remain stable through historical time, then we can be more confident in these interpretations, as well as the effectiveness of subsampling methods in reliably estimating diversity. Recently, this potential issue highlighted by *Jouve et al. (2017)* in a small study of Jurassic and Cretaceous thalattosuchian crocodylomorphs. Those authors tested the conclusions of *Martin et al. (2014)* and their assertion that sea-surface temperature was the primary factor driving marine crocodylomorph evolution, contra *Mannion et al. (2015)* and *Tennant, Mannion & Upchurch (2016a)*. They found that the strength of the relationships reported by the first study, also different to those reported by *Mannion et al. (2015)* and *Tennant, Mannion & Upchurch (2016a)*, were fairly unstable even based on very recent changes in taxonomy. This taxonomically constrained example provides an interesting case of how small changes in publication history can lead to potentially different or conflicting interpretations of macroevolutionary patterns.

In this study, we investigate the influence of publication history on our reading and understanding of diversity patterns through time. For this, we use the clade Dinosauria (excluding Aves) as a study group, as they have an intensely sampled fossil record and a rich history of taxonomic and macroevolutionary research. We note that this is just one of a whole suite of potential biases in palaeodiversity studies (e.g. appropriate time-binning methods, optimal analytical protocols, or the impact of variation in the rock record through space and time), and these factors are discussed in more detail elsewhere (*Peters & Heim, 2010*, *2011b*; *Benson & Butler, 2011*; *Heim & Peters, 2011*; *Benson & Upchurch, 2013*; *Benton et al., 2013*; *Dunhill, Hannisdal & Benton, 2014*; *Benton, 2015*; *Benson et al., 2016*; *Tennant, Mannion & Upchurch, 2016b*).

## MATERIAL AND METHODS

### Dinosaur occurrences dataset

We used a primary dataset of dinosaur body fossil occurrences drawn from the Paleobiology Database (November, 2017) that spans the entirety of the Late Triassic to end-Cretaceous (235–66 Ma) (Supplemental Information 1). These data are based on a comprehensive compilation effort from multiple workers, and represent updated information on modern dinosaur taxonomy and palaeontology at this time. The records comprised only body fossil remains, and excluded ootaxa and ichnotaxa. This dataset was divided into the three major clades, Sauropodomorpha, Ornithischia, and Theropoda. We excluded Aves as they have a fossil record dominated by different and often exceptional modes of preservation. Having limited occurrences of exceptionally preserved fossils will bias our results, particularly in time periods characterised by the presence of avian-bearing Konservat–Lagerstätten (*Brocklehurst et al., 2012*; *Dean, Mannion & Butler, 2016*). We elected to use genera, as these are more readily identified and diagnosed, which means that we can integrate occurrences that are resolved only to the genus level (e.g. *Allosaurus* sp.), and therefore include a substantial volume of data that would be lost at any finer resolution (*Robeck, Maley & Donoghue, 2000*). A potential issue with a genus-level approach is that analysing palaeodiversity at different taxonomic levels can potentially lead to different interpretations about what the external factors

mediating it are (*Wiese, Renaudie & Lazarus, 2016*). Despite the fact that some dinosaur genera are multispecific, it has been shown previously that both genus- and species-level dinosaur diversity curves are very similar (*Barrett, McGowan & Page, 2009*), and that there is more error in species level dinosaur taxonomy than for genera (*Benton, 2008b*). It has also been repeatedly demonstrated that the shape of species and genus curves are strongly correlated in spite of differential taxonomic treatment (*Alroy, 2000a*; *Butler et al., 2011*; *Mannion et al., 2015*), and therefore a genus level compilation should be sufficient for the scope of the present study. We elected to use a stage-level binning method based upon the Standard European Stages and absolute dates provided by *Gradstein et al. (2012)*. Others have used an equal-length time binning approach (*Mannion et al., 2015*; *Benson et al., 2016*), but this has limitations in that it reduces the number of data points for statistical analyses, and can artificially group fossil occurrences from different stages that never temporally co-existed (*Gibert & Escarguel, 2017*), which would confound our analyses. Only body fossil occurrences that could be unambiguously assigned to a single stage bin were included, and those in which assignment to a single stage bin was either ambiguous or not possible were excluded. This procedure was implemented in order to avoid the over-counting of taxa or occurrences that have poorly constrained temporal durations or are contained within multiple time bins. Each dinosaurian sub-group was further sub-divided into approximately contiguous palaeocontinental regions: Africa, Asia, Europe, South America, and North America (*Mannion et al., 2015*). Unfortunately, sampling is too poor to analyse patterns in Antarctica, Australasia, or Indo-Madagascar, although these regions remain included in the global analyses. We also provide data on the number of newly identified occurrences (Supplemental Information 2) and newly named genera (Supplemental Information 3) based on publication date, as well as a list of dinosaur taxa that became invalidated between 1991 and 2015 (Supplemental Information 1).

## Calculating diversity through time

To test how diversity changes through time, we reduced the primary dataset by successively deleting data from publications of each individual occurrence recursively at two-year intervals. Note that these dates are not the same as the date that the actual entries were made into the database, but the explicit date of publication of that occurrence record in the published version of record. We stopped at 1991, giving 12 sequential temporal datasets for each dinosaurian clade. What each version represents is the maturity of the dataset with respect to its present state (and taxonomy as of 2015) based on publication history. Two methods were used to assess diversity patterns. Firstly, empirical diversity based on raw in-bin counts of taxa. This method has been strongly suggested to be a 'biased' or poor estimator of true diversity as it is influenced by heterogeneous sampling (*Benson et al., 2010*; *Benson & Butler, 2011*; *Benson & Upchurch, 2013*; *Butler, Benson & Barrett, 2013*; *Smith & Benson, 2013*; *Newham et al., 2014*; *Mannion et al., 2015*; *Tennant, Mannion & Upchurch, 2016b*). Secondly, we employed the shareholder quorum subsampling (SQS) method, which was designed to account for differences in the shape of

the taxon-abundance curve (*Alroy, 2010a*, *2010c*), and implemented in Perl (Supplemental Informations 4 and 5).

Shareholder quorum subsampling standardises taxonomic occurrence lists based on an estimate of coverage to determine the relative magnitude of taxonomic biodiversity trends (*Alroy, 2010a*, *2010c*). In this method, each taxon within a sample pool (time bin) is treated as a 'shareholder,' whose 'share' is its relative occurrence frequency. Taxa are randomly drawn from compiled in-bin occurrence lists, and when a summed proportion of these 'shares' reaches a certain 'quorum', subsampling stops and the number of sampled taxa is summed. Coverage, as a measure of sampling quality, is defined as the proportion of the frequency distribution of taxa within a sample. It is estimated by using randomized subsampling to calculate the mean value of Good's $u$, which is defined as 1 minus the number of singleton occurrences, divided by the total number of occurrences (*Good, 1953*). A coverage value of zero indicates that all taxa are singleton occurrences (i.e. that all occurrences of a taxon are restricted to a single collection within a time bin). Higher coverage values indicate more even sampling of taxa, and therefore provides a measure of sample completeness that is independent of the overall sample pool size. For each time bin, $u$ is then divided into the quorum level (*Alroy, 2010a*), thereby providing an estimate of the coverage of the total occurrence pool. In all subsampling replicates, singletons were excluded to calculate diversity (but included to calculate Good's $u$), as they can distort estimates of diversity. Dominant taxa (those with the highest frequency of occurrences per bin) were included, and where these taxa are drawn, one is added to the subsampled diversity estimate for that bin (*Alroy, 2010c*). Finally, single large collections that can create the artificial appearance of poor coverage were accounted for by counting occurrences of taxa that only occur in single publications, as opposed to those which occur in single collections, and excluding taxa that are only ever found in the most diverse collection. A total of 1,000 subsampling trials were run for each dataset (Theropoda, Ornithischia, and Sauropodomorpha, for each region and two-year time interval), and the mean diversity was reported for each publication time interval. For each sequential subsampling iteration, whenever a collection from a new publication was drawn from the occurrence list, subsequent collections were sampled until exactly three collections from that publication had been selected (*Alroy, 2010a*). We set a baseline quorum of 0.4, as this has been widely used and demonstrated to be sufficient in accurately assessing changes in diversity (*Alroy, 2010a*, *2010c*; *Mannion et al., 2015*; *Nicholson et al., 2015*; *Tennant, Mannion & Upchurch, 2016a*). Diversity estimates are not reported for any analyses in which this quorum could not be attained. This dual method of using raw and standardised data is important, as not all publications name new taxa; some add to our knowledge of existing taxa by publishing on new occurrences in different collections (or sites). Therefore, by applying a method that accounts for changes in taxonomic abundance across collections we can see how publication history influences diversity through subsampling methods.

## Correlation between diversity extrinsic parameters

For our model-fitting protocol, we follow the procedure outlined in numerous recent analytical studies, by employing simple pairwise correlation tests to the residuals of

detrended time series at the stage level (*Benson & Butler, 2011*; *Butler et al., 2011*; *Butler, Benson & Barrett, 2013*; *Mannion et al., 2015*; *Tennant, Mannion & Upchurch, 2016a*). Residuals for each of the two environmental extrinsic parameters were calculated using the `arima()` function in R, which uses maximum likelihood to fit a first-order autoregressive model to each time series (*Gardner, Harvey & Phillips, 1980*). This method detects the potential influence of any long-term background trend (i.e. a directed change in the mean value of the complete time series through time) within the time series, which has the potential to artificially inflate correlation coefficients in pairwise tests (*Box & Jenkins, 1976*), and also accounts for any potential serial autocorrelation (i.e. the correlation of a variable with itself through successive data points). This protocol has become standard practice now for palaeontological time series analysis following its recommendation by *Alroy (2000a)*. For sea level, we used the curve of *Miller et al. (2005)*, which has been widely applied in recent analyses of tetrapod diversification (*Benson et al., 2010*; *Butler et al., 2011*; *Martin et al., 2014*; *Mannion et al., 2015*; *Tennant, Mannion & Upchurch, 2016a*), and for palaeotemperature we used the data from *Prokoph, Shields & Veizer (2008)*, available as stage level data from *Hannisdal & Peters (2011)* (Supplemental Information 6).

We performed an assessment of normality for each time series prior to any correlation analyses, using the Shapiro–Wilk test (`shapiro.test()` function in R). From the output, if the $p$-values are greater than the pre-defined alpha level (traditionally, 0.05, and used here) this implies that the distribution of the data are not significantly different from a normal distribution, and therefore we can assume normality and use Pearson's test (Pearson's product moment correlation coefficient [$r$]). If $p > 0.05$, we performed a non-parametric Spearman's rank correlation ($\rho$). For each test, both the raw and adjusted $p$-values are reported, the latter calculated using the `p.adjust()` function, and using the 'BH' model (*Benjamini & Hochberg, 1995*). This method accounts for the false-discovery test when performing multiple hypothesis tests with the same data set, which can inflate type-1 error (i.e. in order to avoid falsely rejecting a true null hypothesis; a false positive). We avoided the more commonly used 'Bonferroni correction', due to the undesirable property it has of potentially increasing type-2 error to unacceptable levels (*Nakagawa, 2004*). This adjustment was performed on 'families' of analyses (i.e. non-independent tests), rather than on all correlation tests together, to avoid setting the pass rate for statistical significance too low.

We performed pairwise correlations for the detrended subsampled diversity estimates at each two-year iteration for each group to assess how the strength and direction of correlation changes through publication history. We do not use a maximum likelihood model fitting approach because rather than trying to distinguish between a set of candidate models, we are simply assessing how the strength of correlations changes through publication history.

All analyses were carried out in R version 3.0.2 (*R Development Core Team, 2013*) using the functions available in the default *stats* package.
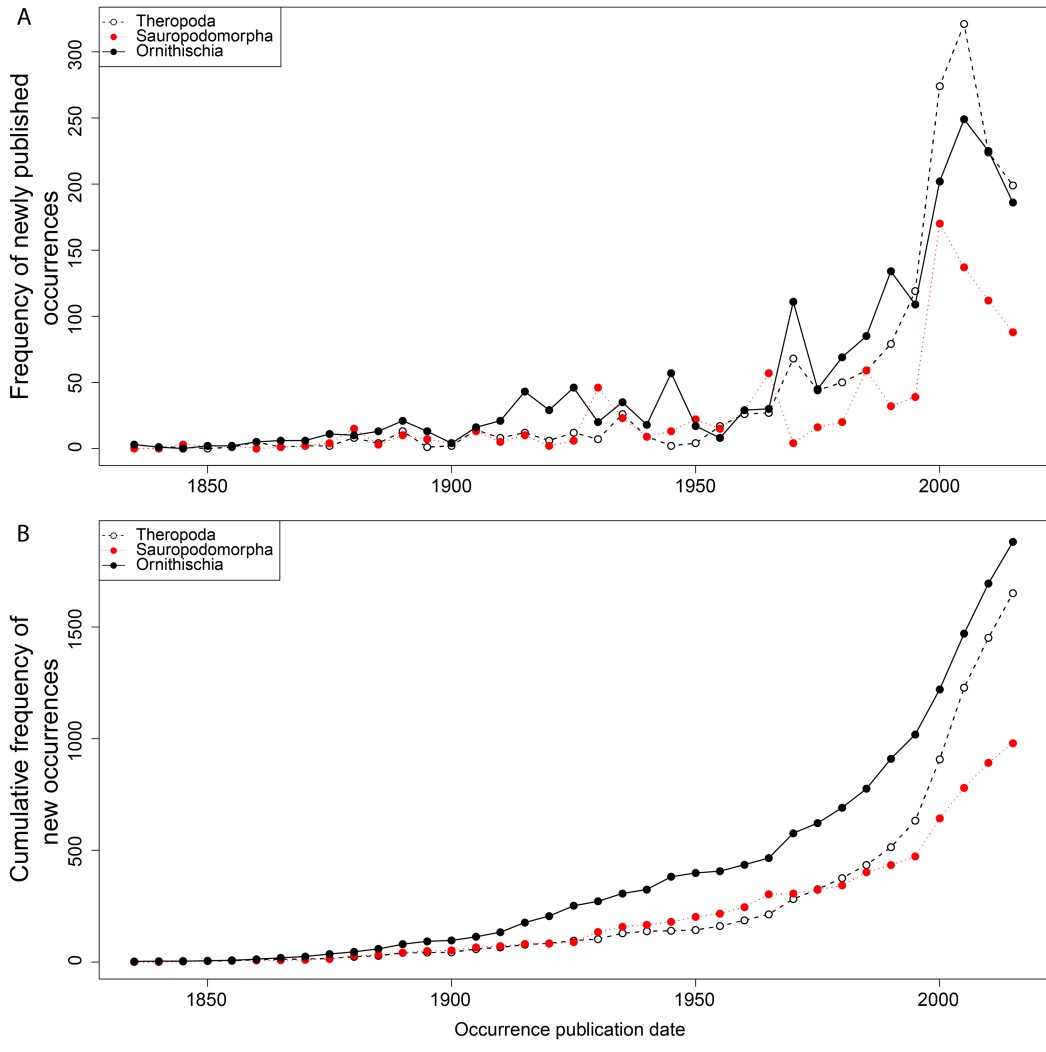

**Figure 1 Frequency (A) and cumulative frequency (B) of newly published dinosaur occurrences through publication time.** Please note that all raw figure files (PDF) and the R code for generating these are available in Supplemental Information 10.            

## RESULTS

### Occurrences and genera through time

From the first dinosaur discoveries until around 1950, the number of dinosaur occurrences published steadily increased through time (Fig. 1). From the mid- to the end of 20th century, the number of published occurrences has increased substantially. This is mostly due to the publication of theropod and ornithischian occurrences, which reached a peak around the turn of the millennium, with occurrences of all three groups remaining high but declining in rate of publication after this. A very similar pattern is observed for genera, with the publication of newly named genera increasing exponentially since around 1990, and at an equal rate for all three groups (Fig. 2). The cumulative frequency of newly named genera shows that, although the rate of growth remains approximately similar and increasing for all three groups, there are times when the relative

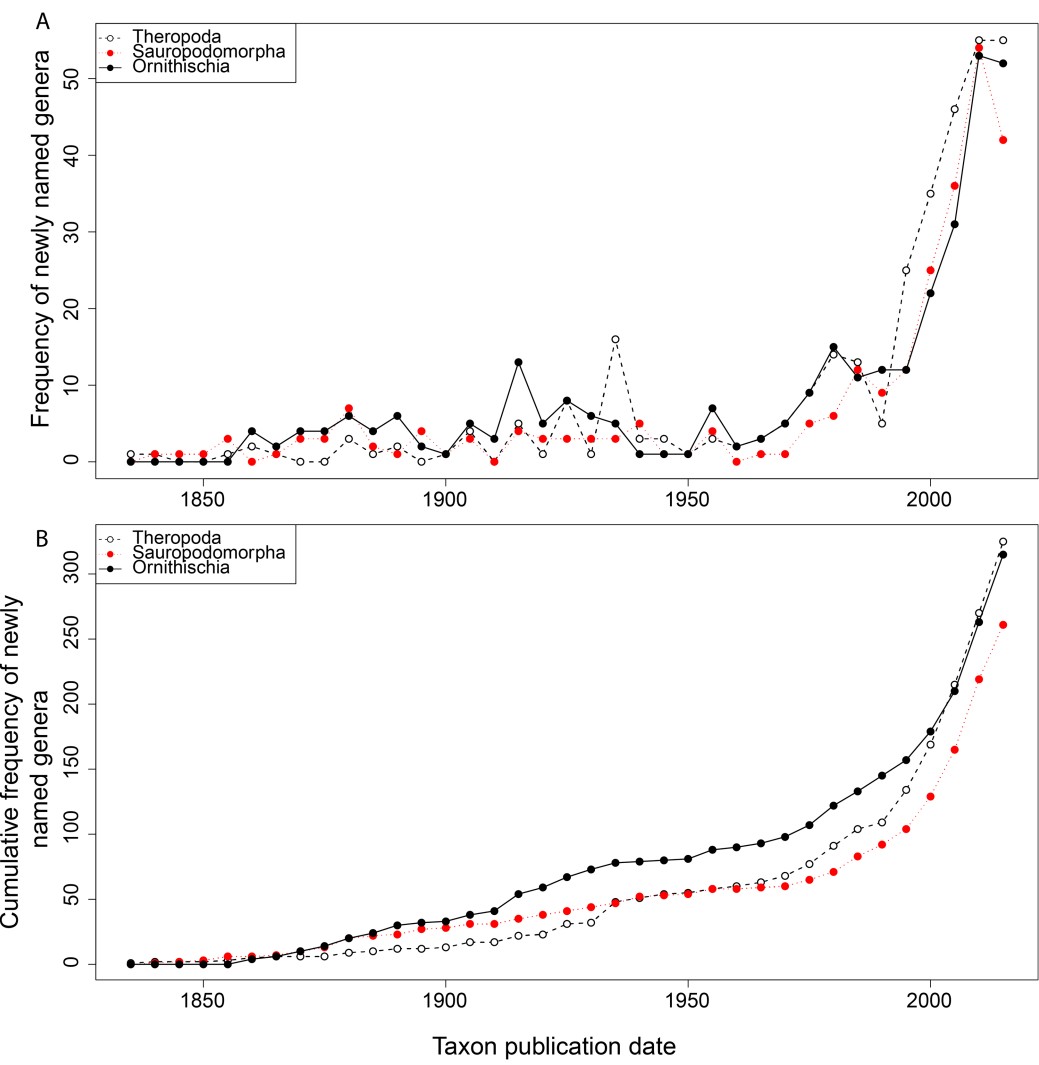

**Figure 2** Frequency (A) and cumulative frequency (B) of newly published dinosaur genera through publication time.

overall number of genera between groups changes through publication history. For example, while sauropodomorphs had more named genera than theropods until around 1935, this changed at around 1960 when new theropod genera became more frequently published than sauropodomorphs. The recent rate of growth of newly named theropod genera in the last 15 years means that they are now named as frequently as newly named ornithischian genera. This recent rate of growth in the naming of new taxa is distinct from the patterns of taxonomic invalidation (e.g. through synonymy) that have occurred since 1991 (Fig. 3). While we see an increase in the number of invalidated taxa between 2000 and 2010, this is variable for each group, with theropods peaking in 2007, sauropodomorphs peaking in 2002–2004, and ornithischians in 2007 and 2013.

### 'Global' patterns of total dinosaur diversity

Apparent 'global' empirical dinosaur diversity steadily rises until the end of the Jurassic (Fig. 4A). Diversity is low across the Jurassic/Cretaceous (J/K) interval until the
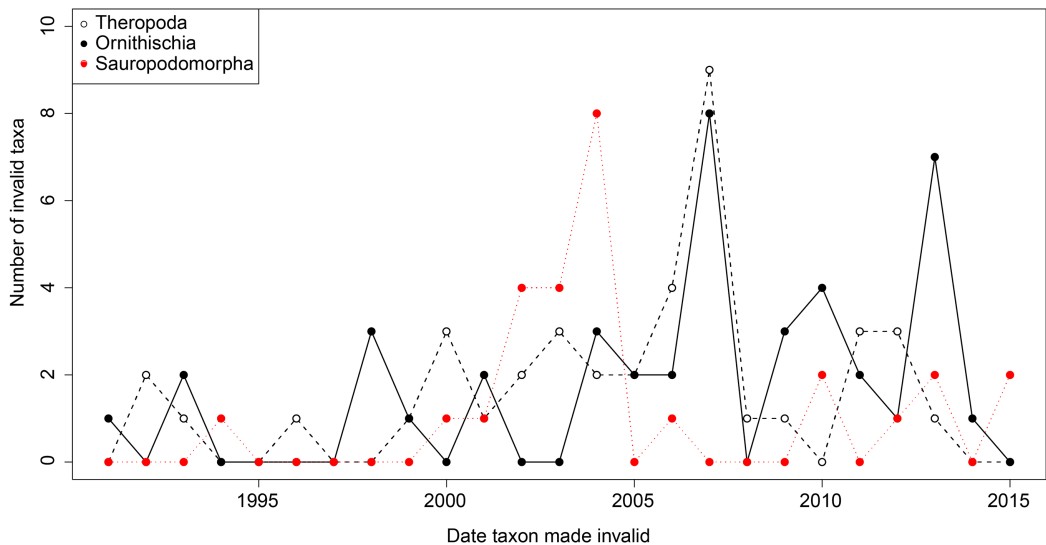

**Figure 3 The number of invalidated or revised dinosaur taxa between 1991 and 2015.**

Hauterivian, before recovering in the late Early Cretaceous. There is a second decline through the late Early to early Late Cretaceous interval, before diversity increases to its zenith in the latest Cretaceous. This general pattern remains constant throughout publication history, although diversity in the 'middle' Cretaceous and latest Cretaceous intervals shows the greatest increases. Subsampled global dinosaur diversity retains this overall pattern (Fig. 4B). The J/K interval decline is still visible, but the late Early Cretaceous apparent diversity increase supersedes Late Jurassic levels. The early Late Cretaceous decline is also still present, but the magnitude of the latest Cretaceous diversity increase is much lower than that recovered for the empirical data. The reason for this distinction between subsampled and raw diversity is that SQS estimates diversity by standardising coverage of the taxon-abundance distribution, and thereby reduces the impact of intensely sampled time intervals such as the latest Cretaceous.

## Patterns of raw and subsampled diversity by group
### Ornithischians

Raw 'global' ornithischian diversity (Fig. 5A) is constant and stable throughout publication history. The apparent magnitude of longer-term trends is obscured by the relative over-sampling of the Campanian and Maastrichtian, which are almost an order of magnitude higher than any other Jurassic or Cretaceous stage interval. Indeed, the Campanian shows no sign of slowing down in increasing diversity, and is the highest and most rapidly increasing of any time interval. In spite of this, the overall trends in raw diversity remain, with steadily increasing Middle to Late Jurassic diversity, a small earliest Cretaceous decline followed by a 'middle' Cretaceous peak in the Aptian, a shallow decline into the early Late Cretaceous, and an increase in the Campanian.

Raw diversity in Europe shows increasing diversity across the J/K transition before an earliest Cretaceous decline (Valanginian–Hauterivian), constant 'middle' Cretaceous

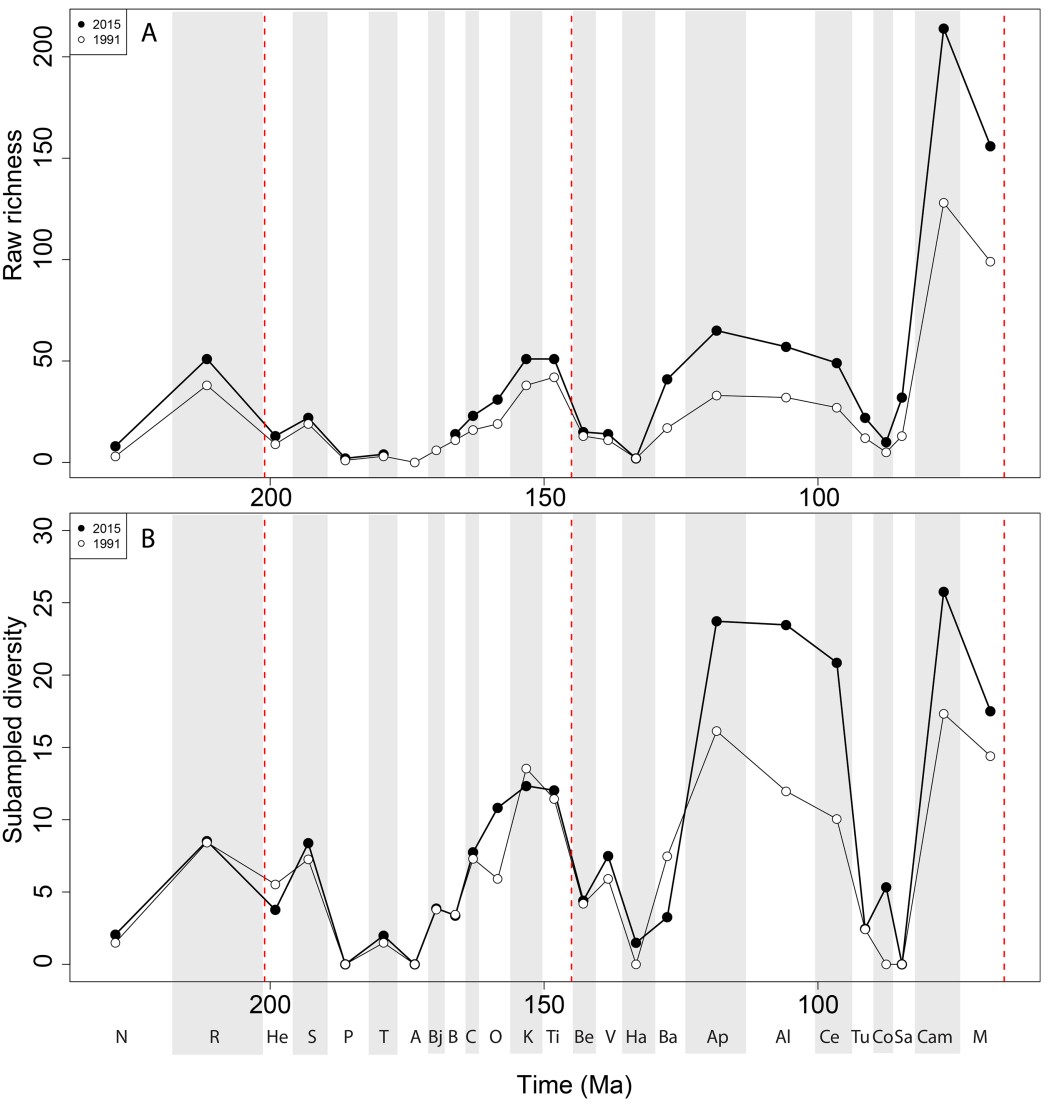

**Figure 4** **Total dinosaur 'global' diversity patterns for (A) raw and (B) subsampled data.** The vertical red lines represent major interval boundaries. Time stage abbreviations (in chronological order). N, Norian; R, Rhaetian, He, Hettangian; S, Sinemurian; P, Pliensbachian; T, Toarcian; A, Aalenian; Bj, Bajocian; B, Bathonian; C, Callovian; O, Oxfordian; K, Kimmeridgian; Ti, Tithonian; Be, Berriasian; V, Valanginian; Ha, Hauterivian; Ba, Barremian; Ap, Aptian; Al, Albian; Ce, Cenomanian; Tu, Turonian; Co, Coniacian; Sa, Santonian; Cam, Campanian; M, Maastrichtian. Vertical dashed red lines indicate boundaries between different periods (Triassic/Jurassic, Jurassic/Cretaceous, and Cretaceous/Paleogene).

diversity, and an increase from the Campanian to Maastrichtian (Fig. 5B). Raw African ornithischian diversity is too inconsistent to analyse any changes through geological time or publication time (Fig. 5C). Raw Asian diversity is fairly constant through the Cretaceous, until an apparent major Campanian peak and Maastrichtian decline (Fig. 5D). In North America, empirical diversity is flat and low throughout the Late Jurassic and most of the Cretaceous (Fig. 5E). There is a Campanian peak, and order of magnitude higher than any prior interval, which is rapidly increasing through publication time.

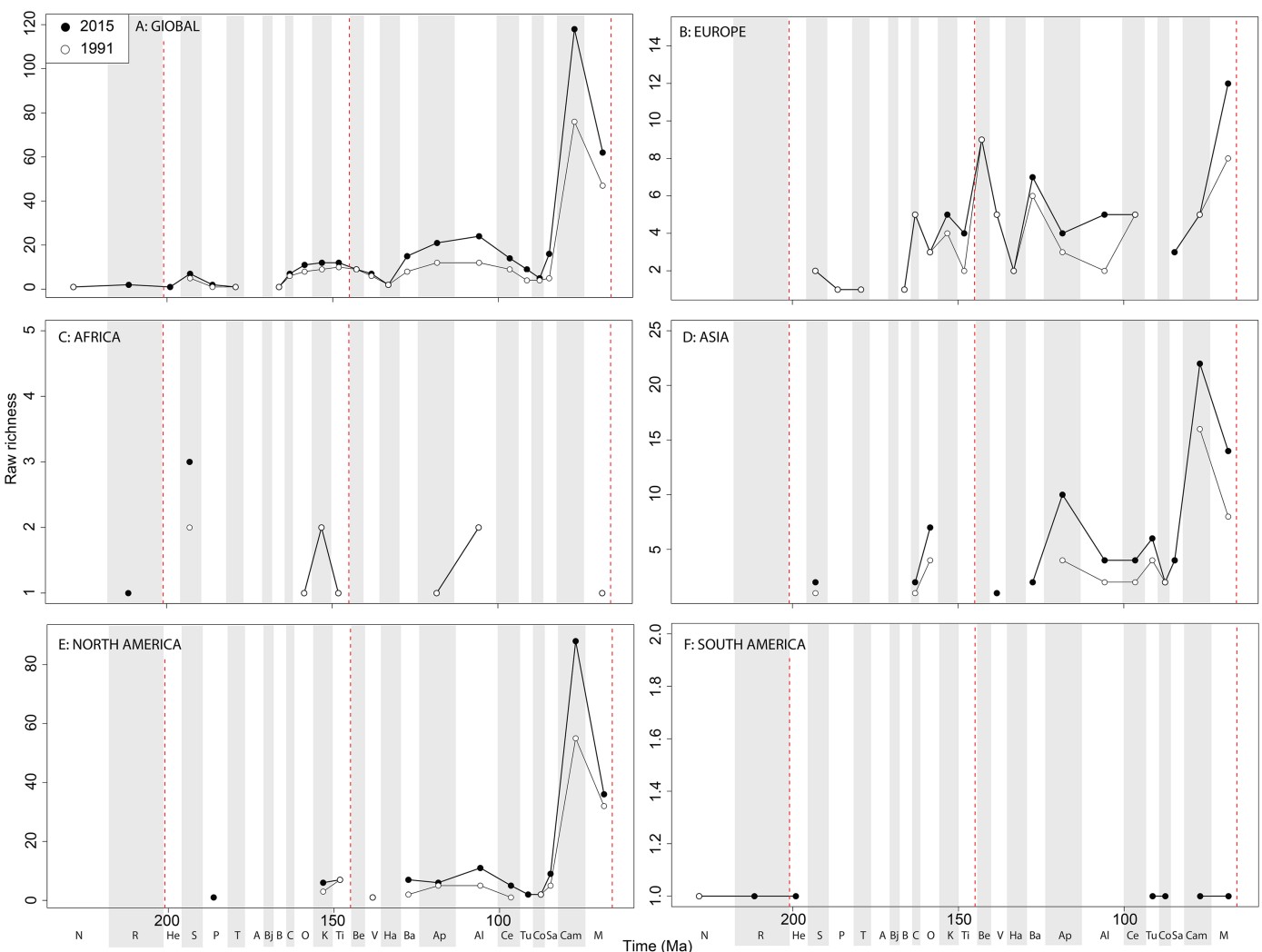

**Figure 5 Raw ornithischian diversity at (A) global and (B–F) regional levels (Europe, Africa, Asia, North America, and South America, respectively) based on our published knowledge in 1991 and 2015.** Abbreviations as Fig. 4.

Diversity decreases from this into the Maastrichtian, in which diversity has remained relatively stable through publication time. Sampling in South America is also relatively poor, with apparent diversity remaining low and flat where a signal is obtained (Fig. 5F).

Subsampled 'global' ornithischian diversity shows a distinctly different pattern from the raw curve, both in terms of overall trends, and in terms of the magnitude of the effect of publication history (Fig. 6A). The Jurassic is generally too poorly sampled to reveal a constant signal, but there is evidence of a decline through the J/K transition, which remains constant through publication time. This is followed by a middle-Cretaceous increase, in which ornithischian diversity is at its second highest level throughout their history. The magnitude of this Albian radiation has rapidly increased over publication time, the result being that originally what appeared to be increasing subsampled diversity over the Early/Late Cretaceous transition now shows a major decline from the Albian to Coniacian. Santonian subsampled diversity remains unknown, but when we see a signal

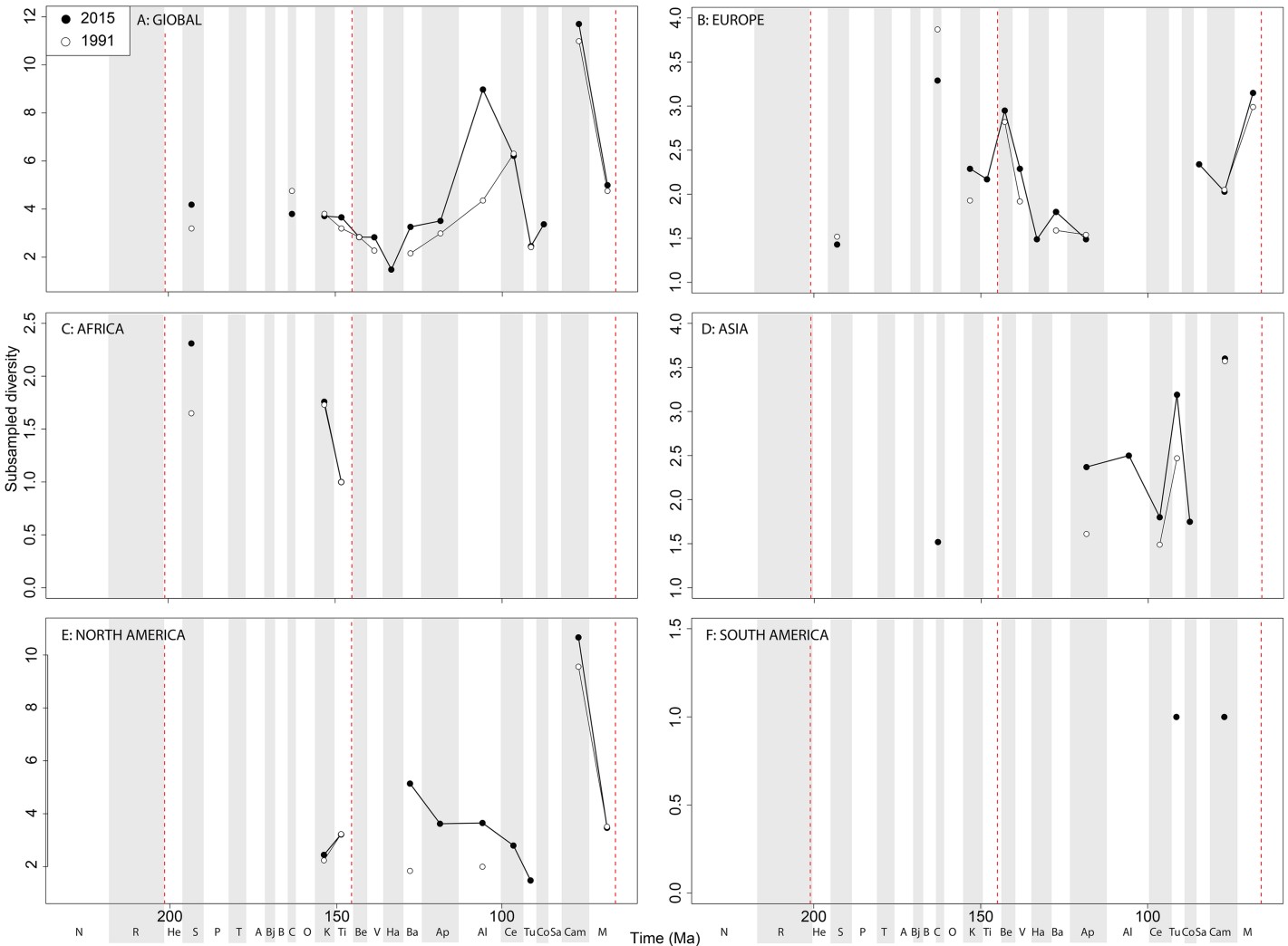

**Figure 6 Subsampled ornithischian diversity at (A) global and (B–F) regional levels (Europe, Africa, Asia, North America, and South America, respectively) based on our published knowledge in 1991 and 2015.** Abbreviations as Fig. 4.

emerge in the Campanian, diversity is higher than the Albian, reaching its highest level before declining by more than half into the Maastrichtian. This overall structure, besides the Albian, remains consistent throughout publication time with no major perturbations to the apparent 'global' curve.

Subsampled European diversity reveals increasing diversity across the Tithonian/Berriasian transition, followed by overall gradually decreasing diversity throughout the remainder of the Early Cretaceous (Fig. 6B). In Africa, the signal is too poor to reveal anything besides a Kimmeridgian/Tithonian subsampled diversity drop (Fig. 6C), and in Asia, there is evidence of a decline in subsampled diversity across the Albian/Cenomanian transition (Fig. 6D). In North America, subsampled diversity reveals a decline across the Early–Late Cretaceous transition, and a major decline from the Campanian to Maastrichtian, a pattern that remains stable through publication history (Fig. 6E). In South America, the subsampled signal is too poor to comment on ornithischian diversity (Fig. 6F).

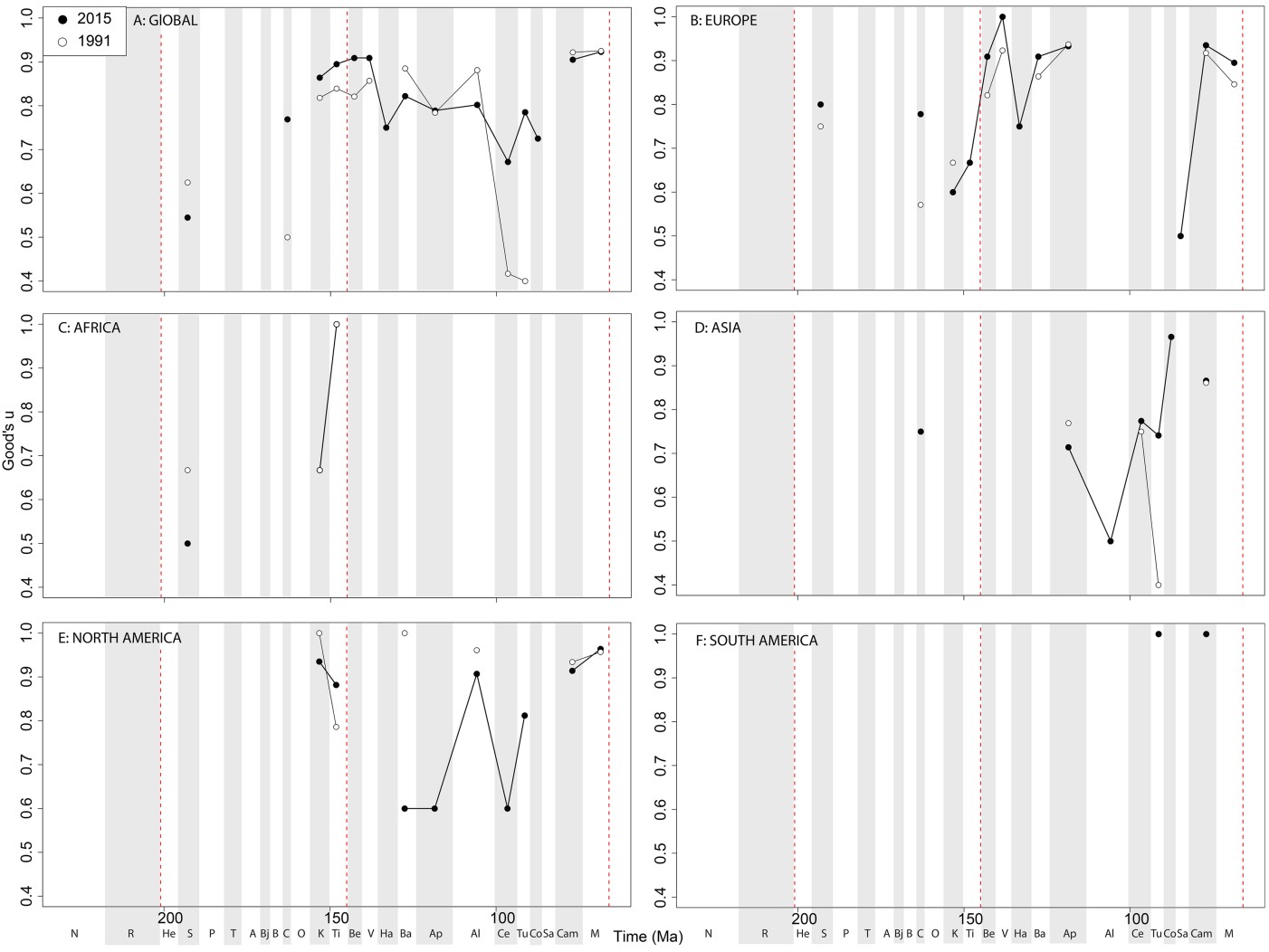

**Figure 7 Good's *u* estimates for ornithischians at (A) global and (B–F) regional levels (Europe, Africa, Asia, North America, and South America, respectively) based on our published knowledge in 1991 and 2015.** Abbreviations as Fig. 4.

If we look at how coverage has changed through publication history (based on Good's *u*), we should expect that subsampled diversity patterns are reflective of this pattern. At a global level, coverage in the Cretaceous is much better than the Jurassic (Fig. 7A). Much of this, however, is based on patchy regional records. In Europe, we find that coverage increases across the J/K interval (Fig. 7B), and is the only place where a consistently reliable record here can be obtained. In Africa, coverage is generally poor, besides in the latest Jurassic (Fig. 7C). In Asia, coverage is poor up until the late Early Cretaceous (Fig. 7D). In North America, coverage is good in the latest Jurassic and 'middle' to Late Cretaceous, but non-existent in Early to Middle Jurassic and earliest Cretaceous (Fig. 7E). Coverage is generally poor for the entire South American ornithischian record (Fig. 7F), explaining why obtaining a subsampled diversity signal here is difficult.

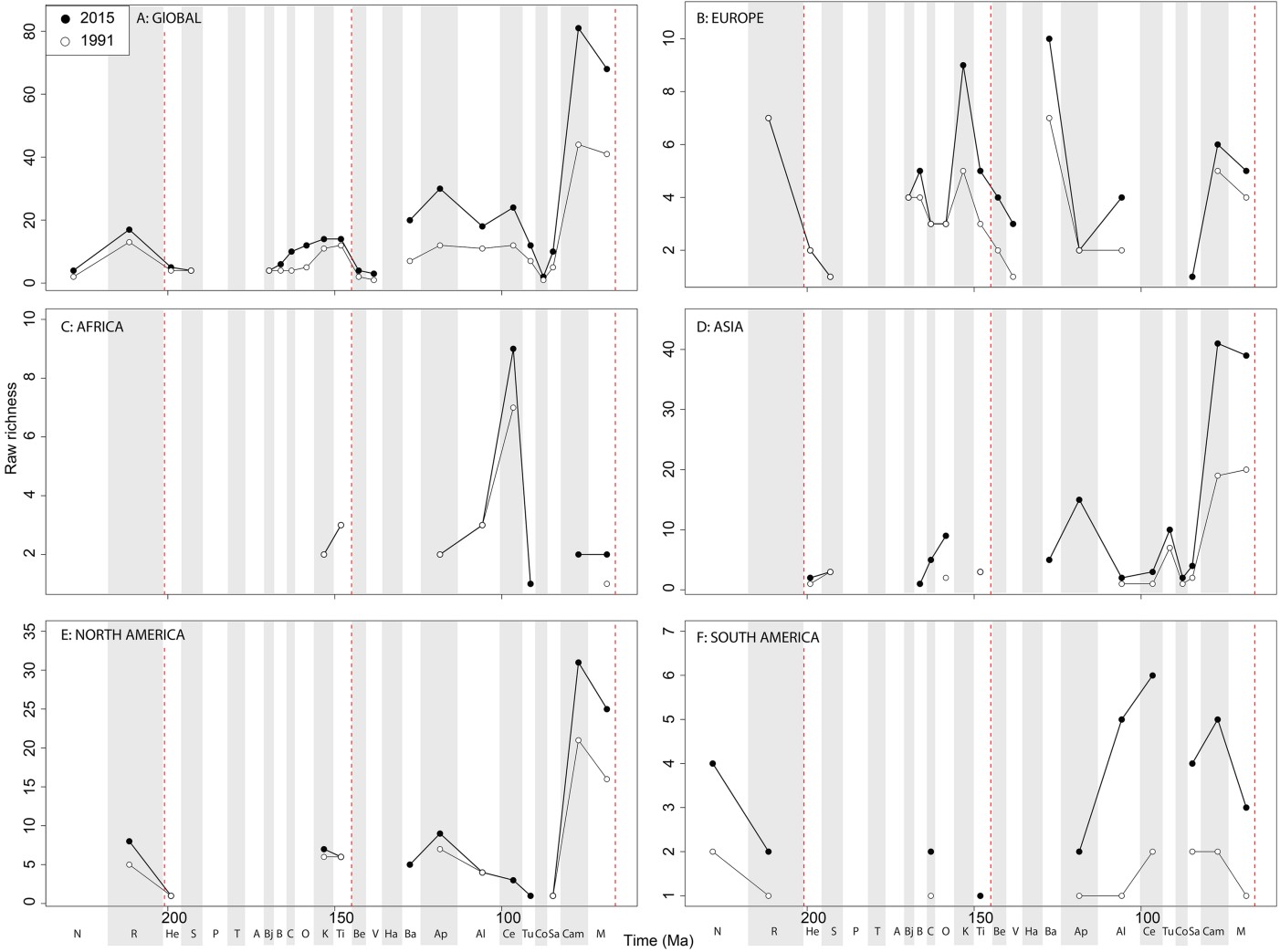

**Figure 8 Raw theropod diversity at (A) global and (B–F) regional levels (Europe, Africa, Asia, North America, and South America, respectively) based on our published knowledge in 1991 and 2015.** Abbreviations as Fig. 4.

## Theropods

The overall shape of the raw 'global' theropod diversity curve remains consistent through publication history for the Jurassic (Fig. 8A), similar to ornithischians, where we see steadily increasing Middle–Late Jurassic diversity. 'Middle' Cretaceous raw diversity fluctuated, followed by a major Campanian–Maastrichtian rise. The lowest apparent diversity is in the Coniacian, reaching earliest Cretaceous levels. Notable variations due to publication history are in the Barremian–Cenomanian, where diversity increases in magnitude through time, gradually exceeding that for Late Jurassic diversity. Raw European diversity is fairly constant through publication history (Fig. 8B), with a Middle Jurassic diversity peak in the Bathonian, followed by a Callovian–Oxfordian trough, a second larger Kimmeridgian peak, and then constant decline from the Tithonian to the Valanginian. Barremian diversity is increases through publication time, and is as high as Kimmeridgian levels. Aptian and Albian diversity is relatively low through publication

history. Campanian and Maastrichtian diversity levels are slowly increasing through publication history. As with ornithischians, African theropods are generally too poorly sampled at the stage level to recognise any consistent empirical patterns (Fig. 8C). There is a Cenomanian raw diversity spike, but how this compares with much of the rest of the Cretaceous is obscured by patchy sampling. In Asia, raw Late Jurassic diversity is generally lower than for the Cretaceous (Fig. 8D). The Cretaceous sees three peaks in apparent diversity during the Aptian, Turonian, and Campanian–Maastrichtian, with the latter being considerably higher than any previous one, and growing rapidly through publication history. In North America, raw diversity levels are dwarfed by the intensive sampling of latest Cretaceous theropods, with major gaps in the Middle–Late Jurassic and earliest Cretaceous records (Fig. 8E). Campanian and Maastrichtian raw diversity is constantly increasing at a faster rate than any other time interval, and consistently reveals a slight apparent diversity decline into the end-Cretaceous. Raw South American diversity estimates are changing rapidly through publication history, with almost every interval in which dinosaurs are available to be sampled doubling or tripling since 1991 (Fig. 8F). Of note is a recently emerging Late Jurassic theropod fossil record in South America, which at the present reveals an apparent low diversity.

When subsampling is applied, in the Late Jurassic we see a switch from steadily increasing subsampled diversity to a major Oxfordian peak and subsequent decline in diversity through the J/K transition decline, a pattern that is consistently recovered through publication time (Fig. 9A). Subsampled diversity is at its highest level during the Aptian than at any other stage during theropod history, and has doubled in the last 20 years of publication history. Campanian and Maastrichtian diversity are as high as the Cenomanian, a pattern that remains consistent through publication time. We see the 'global' J/K transition decline reflected in Europe (Fig. 9B), and a strong Barremian peak, which is not captured on a 'global' scale. Latest Triassic subsampled diversity is higher than at any other point in the Jurassic in Europe. Maastrichtian subsampled diversity remains high, reaching the same level as that for the Kimmeridgian. In Africa, as with ornithischians the signal is very patchy after subsampling is applied (Fig. 9C), but captures an Albian–Cenomanian diversity increase, which remains constant throughout publication history, and flat diversity in the latest Cretaceous. The subsampled theropod diversity signal is also patchy in Asia, but does reveal a very high latest Cretaceous diversity level, which is not otherwise seen throughout theropod evolutionary history (Fig. 9D). In North America, the subsampled record is as patchy as that for ornithischians, but remains stable through publication history (Fig. 9E). Here, we see slightly increasing subsampled diversity in the latest Jurassic, a large decline from the Aptian to Albian, and a major diversification from the Santonian to Campanian. In South America, a subsampled diversity signal is almost entirely absent, although we do see a reduction in almost half from the Norian to Rhaetian, which remains stable through publication history (Fig. 9F).

Theropod coverage levels are quite patchy at the 'global' level, remaining constant in the Late Triassic, fluctuating in the Middle Jurassic to earliest Cretaceous, but remaining fairly stable in the 'middle' and latest Cretaceous through publication history (Fig. 10A). On a regional level, this apparent 'global' signal across the J/K transition is again

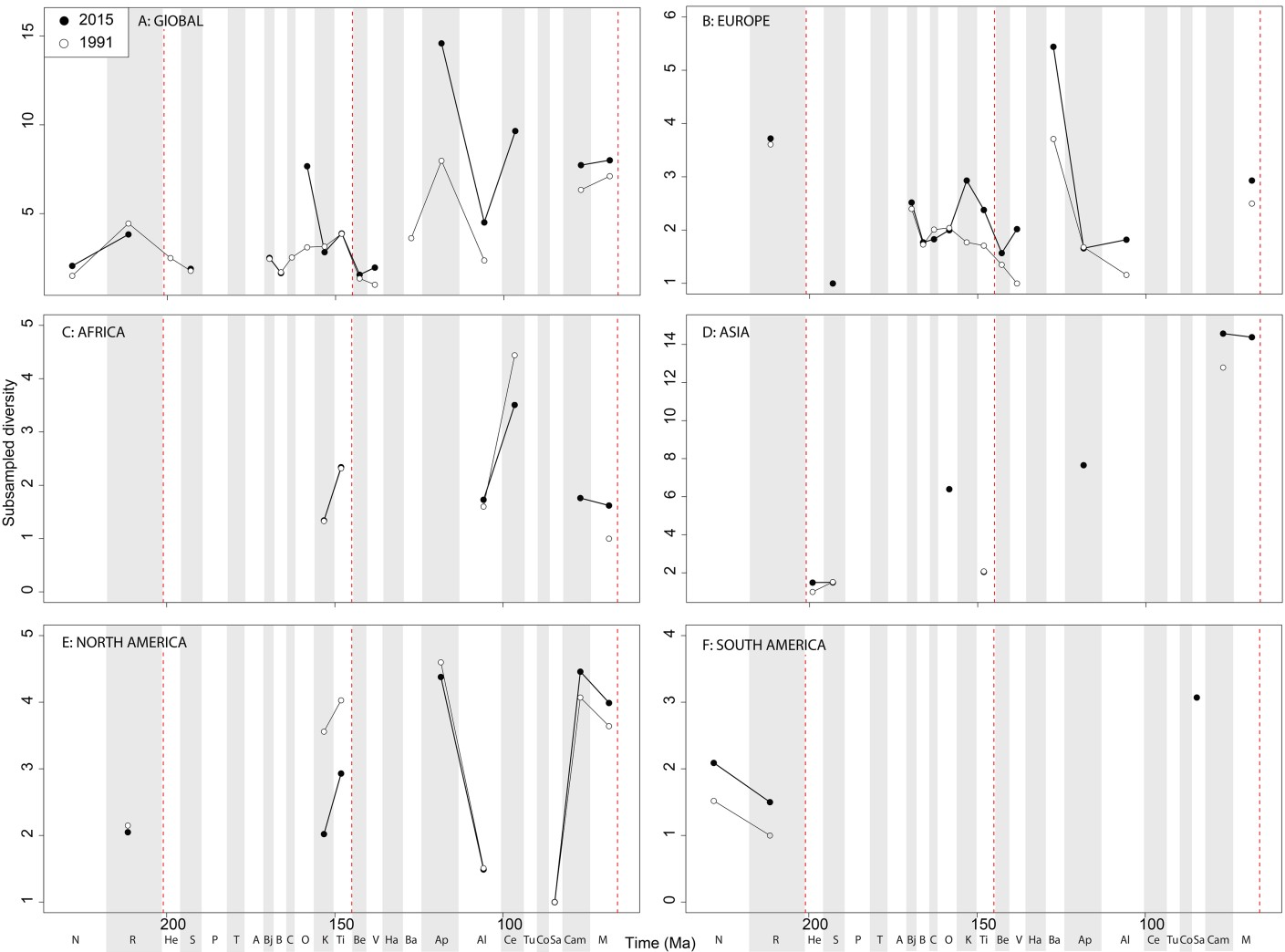

**Figure 9  Subsampled theropod diversity at (A) global and (B–F) regional levels (Europe, Africa, Asia, North America, and South America, respectively) based on our published knowledge in 1991 and 2015.** Abbreviations as Fig. 4.

emphasised in Europe, but in the Valanginian and Albian, coverage is getting notably worse through publication history (Fig. 10B). Coverage in Africa (Fig. 10C) and Asia (Fig. 10D) is very patchy, and does not appear to have changed in the last 20 years overall, besides the origin of moderate coverage levels in the Oxfordian and Aptian of Asia. In North America, coverage levels are moderately high in the latest Jurassic, Aptian, and Albian, and latest Cretaceous, only improving in the latest Jurassic through publication history (Fig. 10E). In South America, coverage is generally poor throughout the Jurassic and Cretaceous, but appears to be declining in the Norian and Rhaetian theropod records (Fig. 10F).

### *Sauropodomorphs*
Sauropodomorph empirical diversity emphasises some more changes in raw patterns through publication time, particularly in the 'middle' and Late Cretaceous (Fig. 11A).

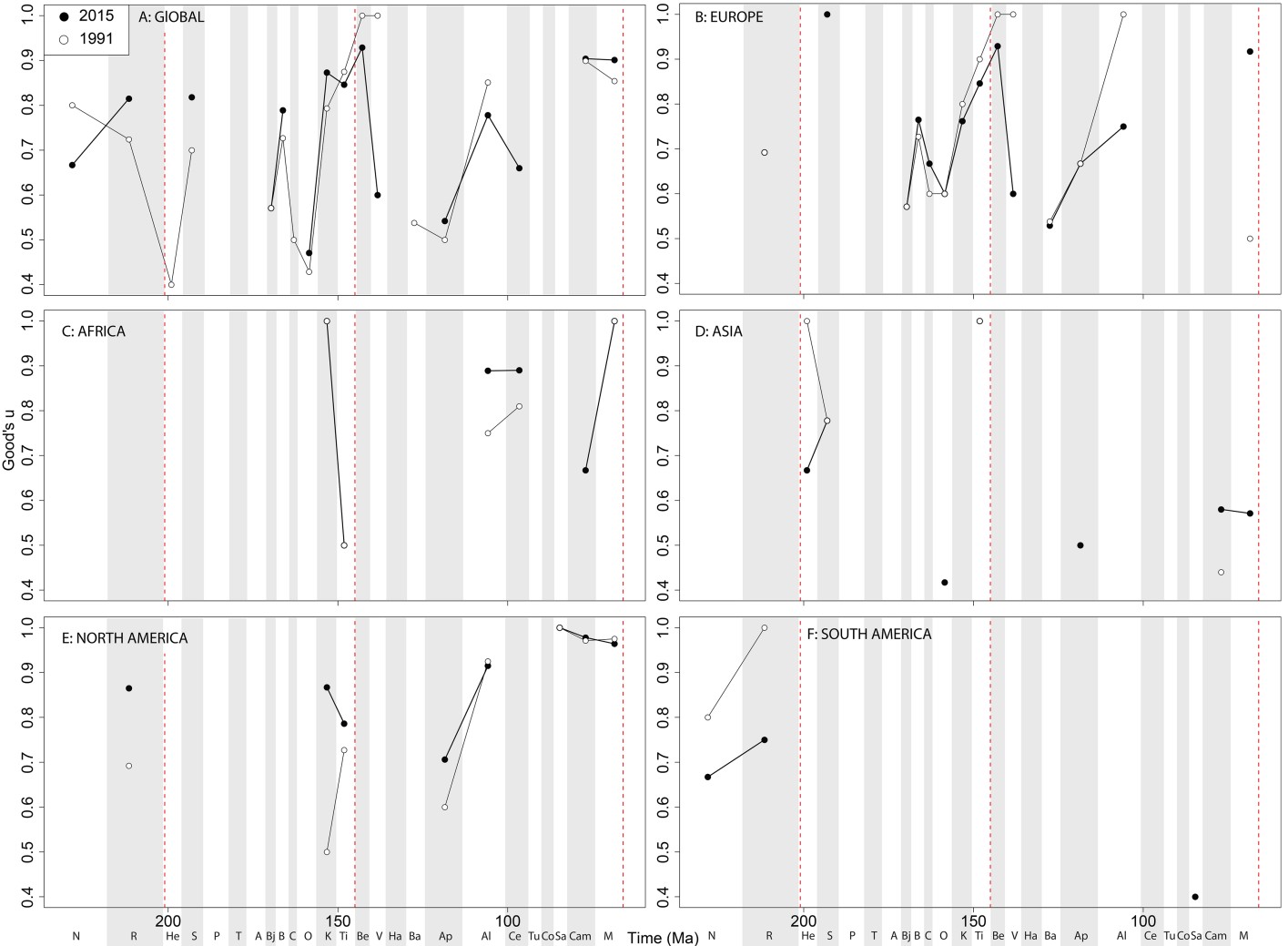

**Figure 10  Good's _u_ estimates for theropods at (A) global and (B–F) regional levels (Europe, Africa, Asia, North America, and South America, respectively) based on our published knowledge in 1991 and 2015.** Abbreviations as Fig. 4.     

Late Jurassic patterns are fairly consistent, with a rising Kimmeridgian and Tithonian raw diversity emphasising an apparent major decline across the J/K interval. In Europe, sauropods show a consistent and major decline in raw diversity from the Kimmeridgian to the Berriasian (Fig. 11B). Much of the rest of the Cretaceous is too poorly sampled, but raw sauropod diversity never attains Kimmeridgian levels in Europe for the rest of their evolutionary history. Sauropodomorph dinosaurs are generally better sampled than theropods and ornithischians in Africa, showing an apparent decline through the Triassic/Jurassic transition, a latest Jurassic raw diversity peak, and low levels through the 'middle' to Late Cretaceous transition (Fig. 11C). In Asia, raw taxonomic diversity is generally low compared to the Maastrichtian, in which diversity is relatively high and still rapidly increasing through publication history (Fig. 11D). The North American sauropod record is very patchy, with the latest Jurassic showing a shift from rapidly increasing raw diversity from the Oxfordian to a slight drop from the Kimmeridgian to Tithonian (Fig. 11E).

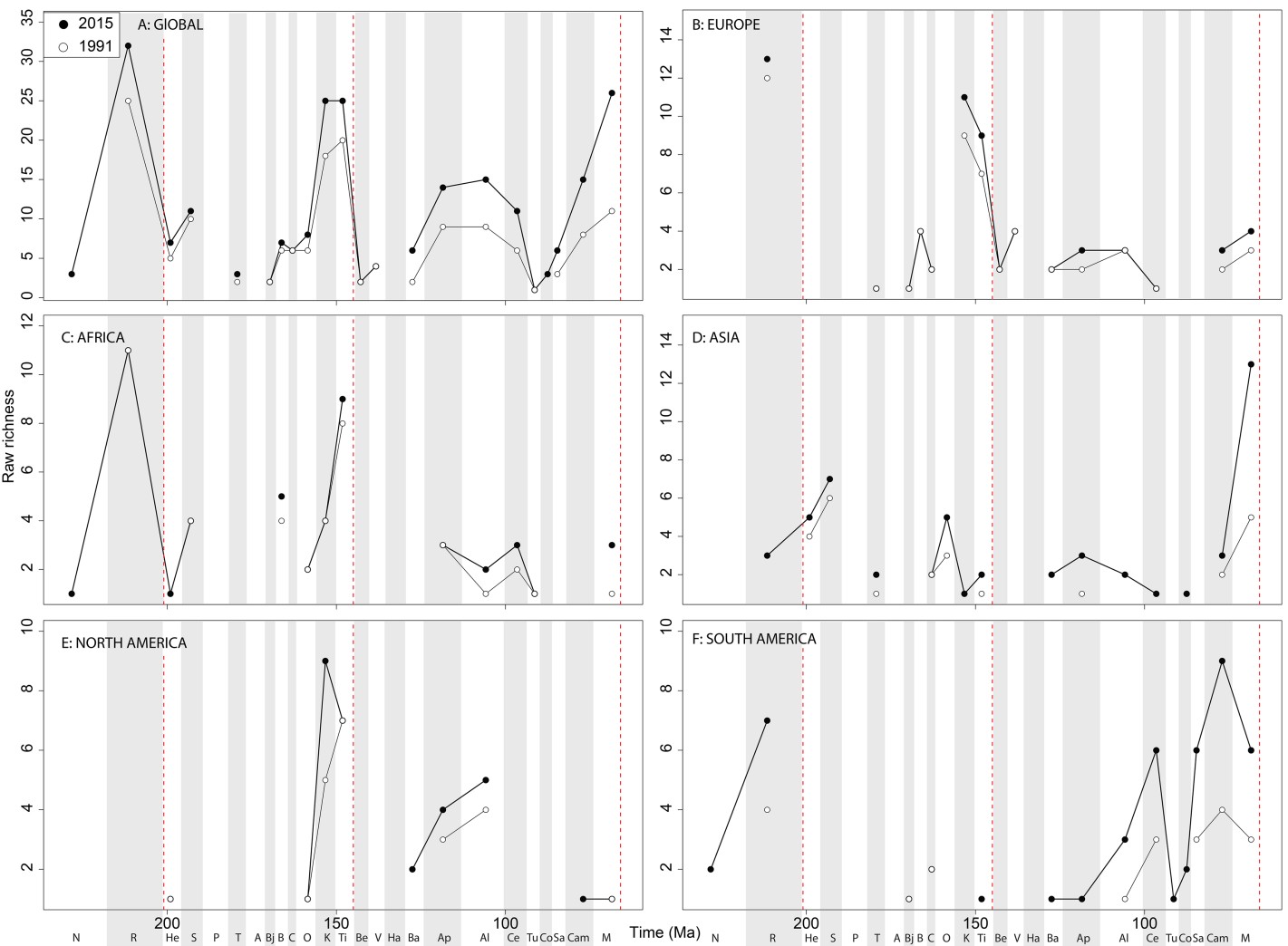

**Figure 11 Raw sauropodomorph diversity at (A) global and (B–F) regional levels (Europe, Africa, Asia, North America, and South America, respectively) based on our published knowledge in 1991 and 2015.** Abbreviations as Fig. 4.

The South American Jurassic sauropod record is patchy, but raw diversity is increasing throughout the 'middle' to Late Cretaceous through publication history (Fig. 11F).

At a 'global' level, Jurassic sauropodomorph subsampled diversity remains consistent through publication history (Fig. 12A). Here, we see steadily increasing diversity levels through the Middle and Late Jurassic, before a decline through the J/K transition, which might have been initiated before the J/K boundary itself. The greatest change in subsampled diversity is in the Albian, which has almost doubled in the last 20 years, with implication for the 'mid-Cretaceous sauropod hiatus' (*Mannion & Upchurch, 2011*). Subsampling reduces the European diversity signal due to poor sampling of sauropods, although there is evidence for the sauropod decline beginning prior to the J/K transition (Fig. 12B). In Africa, when subsampling is applied, the few intervals in which a signal emerges reveal a fairly constant level of diversity through the Jurassic and Cretaceous, and through publication time, with the notable exception being an increase in subsampled

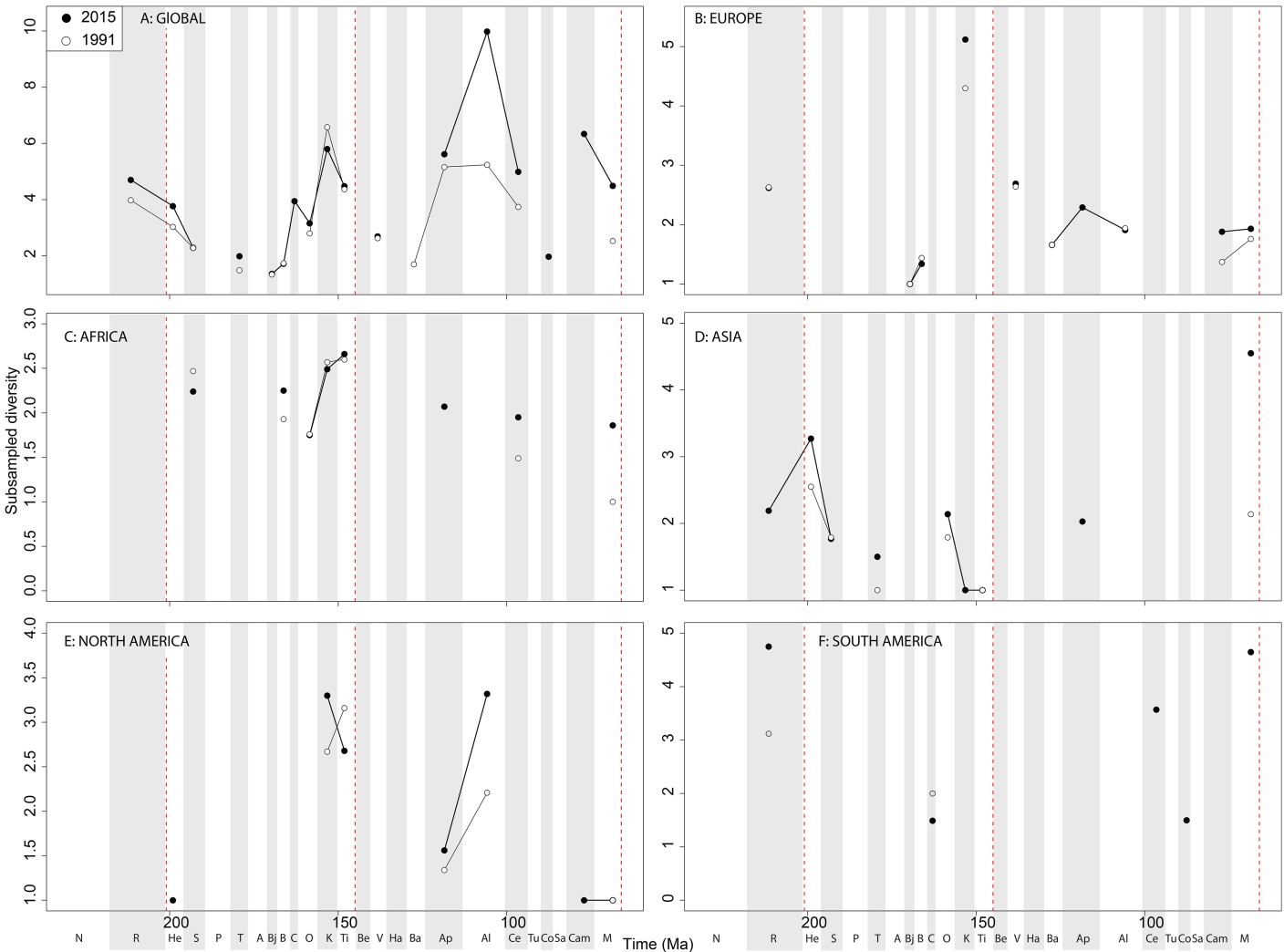

**Figure 12 Subsampled sauropodomorph diversity at (A) global and (B–F) regional levels (Europe, Africa, Asia, North America, and South America, respectively) based on our published knowledge in 1991 and 2015.** Abbreviations as Fig. 4.

diversity in the latest Jurassic (Fig. 12C). In Asia, the signal is also fairly poor after subsampling is applied (Fig. 12D). Here, we see an increase in subsampled diversity across the Triassic/Jurassic transition, and the highest diversity level is in the Maastrichtian, where subsampled estimates have increased by more than double in the last 20 years. In North America, the subsampled signal is highly degraded, although of note is a near doubling of Albian diversity levels in the last 20 years (Fig. 12E). In South America, the signal is very inconsistent, but improving through publication history, with a patchy Late Cretaceous signal beginning to emerge (Fig. 12F). Full subsampling results are provided in Supplemental Informations 7 and 8.

Sauropodomorph coverage varies greatly at the 'global' level, with high levels in the Triassic-Jurassic transition, the Middle and Late Jurassic (with the exception of the Callovian), and the Maastrichtian (Fig. 13A). As with theropods and ornithischians,

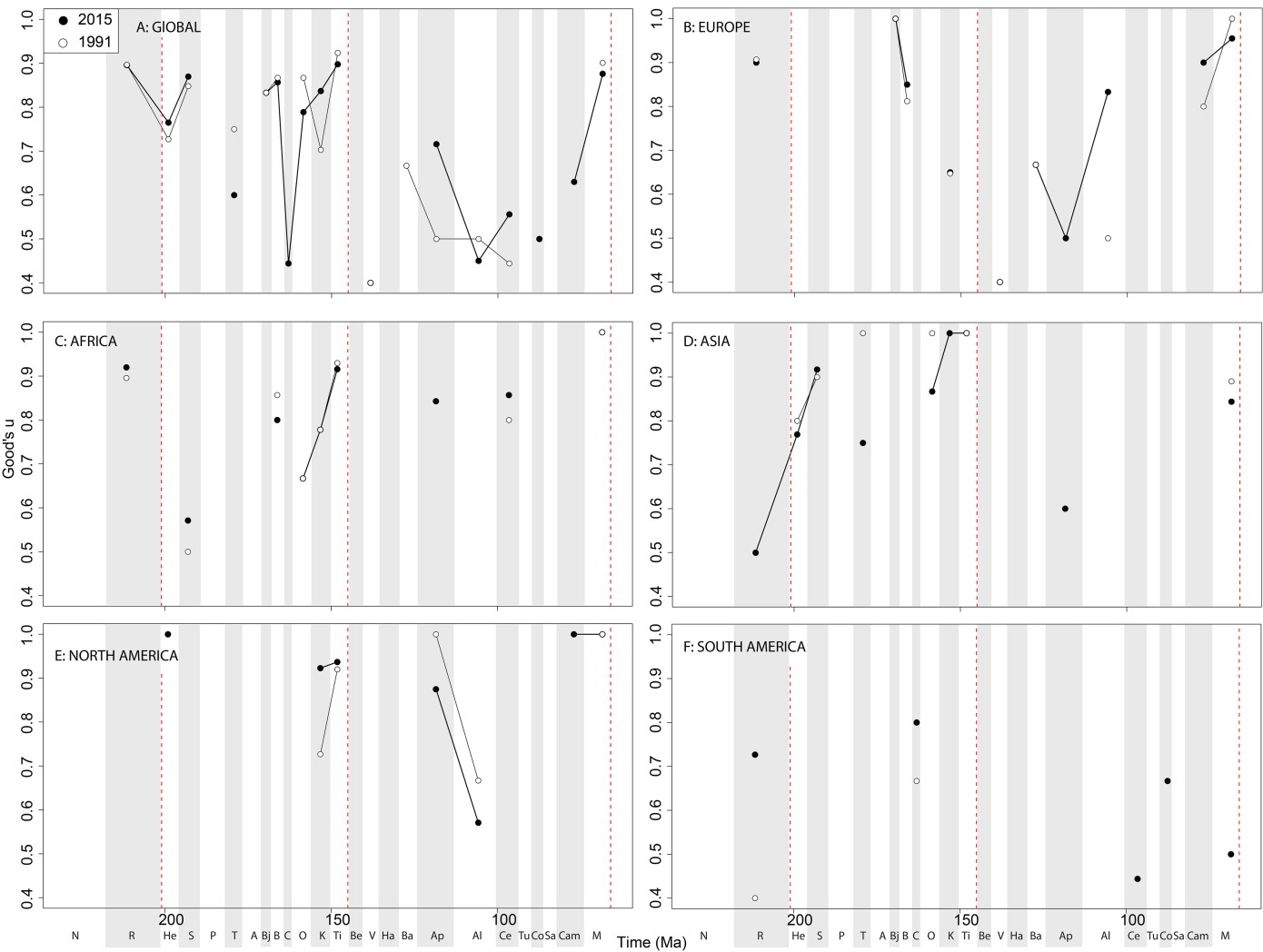

**Figure 13** Good's *u* estimates for sauropodomorphs at a (A) global and (B–F) regional levels (Europe, Africa, Asia, North America, and South America, respectively) based on our published knowledge in 1991 and 2015. Abbreviations as Fig. 4.

however, this is a composite of a very patchy regional record. In Europe, coverage is high during the latest Triassic, Middle Jurassic, and Late Cretaceous, and this does not seem to have varied with publication time (Fig. 13B). In Africa, moderate levels of coverage also have not changed substantially since 1991 (Fig. 13C). In Asia, coverage is generally high in the Late Jurassic, but the Cretaceous record is incredibly poor with just two data points (Aptian and Maastrichtian; Fig. 13D). In North America, the latest Jurassic has high coverage levels, which are increasing through publication history in the Kimmeridgian, and moderately high coverage in the Aptian and latest Cretaceous (Fig. 13E). In South America, coverage is very patchy and inconsistent, with the only noteworthy change through publication history being an increase for the Rhaetian interval (Fig. 13F).

## Correlation results

Our results find varying strength of correlation between subsampled 'global' dinosaur diversity for each clade and both palaeotemperature and sea level, although the

**Table 1 Ornithischian correlation test results.**

| Ornithischians | | | Sea level | | | Palaeotemperature | | |
|---|---|---|---|---|---|---|---|---|
| | Shapiro–Wilk (p) | Correlation test | cor | p | Adjusted p | cor | p | Adjusted p |
| 2015 | 0.003 | Spearman | 0.42 | 0.137 | 0.322 | −0.432 | 0.109 | 0.235 |
| 2013 | 0.002 | Spearman | 0.481 | 0.084 | 0.273 | −0.396 | 0.145 | 0.235 |
| 2011 | 0.002 | Spearman | 0.481 | 0.084 | 0.273 | −0.396 | 0.145 | 0.235 |
| 2009 | 0.002 | Spearman | 0.516 | 0.062 | 0.273 | −0.429 | 0.113 | 0.235 |
| 2007 | 0.001 | Spearman | 0.503 | 0.069 | 0.273 | −0.471 | 0.078 | 0.235 |
| 2005 | <0.001 | Spearman | 0.358 | 0.209 | 0.273 | −0.346 | 0.206 | 0.237 |
| 2003 | 0.002 | Spearman | 0.314 | 0.274 | 0.322 | −0.325 | 0.237 | 0.237 |
| 2001 | 0.001 | Spearman | 0.332 | 0.246 | 0.322 | −0.329 | 0.232 | 0.237 |
| 1999 | 0.002 | Spearman | 0.327 | 0.253 | 0.322 | −0.432 | 0.109 | 0.235 |
| 1997 | 0.001 | Spearman | 0.341 | 0.233 | 0.322 | −0.429 | 0.113 | 0.235 |
| 1995 | <0.001 | Spearman | 0.258 | 0.394 | 0.394 | −0.367 | 0.197 | 0.237 |
| 1993 | 0.001 | Spearman | 0.413 | 0.185 | 0.322 | −0.495 | 0.089 | 0.235 |
| 1991 | 0.002 | Spearman | 0.329 | 0.297 | 0.322 | −412 | 0.163 | 0.235 |

**Table 2 Sauropodomorph correlation test results.**

| Sauropodomorphs | | | Sea level | | | Palaeotemperature | | |
|---|---|---|---|---|---|---|---|---|
| | Shapiro–Wilk (p) | Correlation test | cor | p | Adjusted p | cor | p | Adjusted p |
| 2015 | 0.036 | Spearman | −0.114 | 0.711 | 0.795 | −0.171 | 0.527 | 0.609 |
| 2013 | 0.045 | Spearman | −0.08 | 0.795 | 0.795 | −0.138 | 0.609 | 0.609 |
| 2011 | 0.274 | Pearson | 0.399 | 0.201 | 0.877 | 0.095 | 0.736 | 0.81 |
| 2009 | 0.192 | Pearson | 0.399 | 0.201 | 0.877 | 0.067 | 0.813 | 0.813 |
| 2007 | 0.052 | Pearson | 0.161 | 0.619 | 0.877 | −0.197 | 0.482 | 0.81 |
| 2005 | 0.477 | Pearson | 0.115 | 0.71 | 0.877 | −0.221 | 0.41 | 0.81 |
| 2003 | 0.19 | Pearson | 0.168 | 0.614 | 0.877 | −0.235 | 0.4 | 0.81 |
| 2001 | 0.385 | Pearson | 0.007 | 0.991 | 0.991 | −0.199 | 0.477 | 0.81 |
| 1999 | 0.124 | Pearson | 0.105 | 0.75 | 0.877 | −0.174 | 0.522 | 0.81 |
| 1997 | 0.887 | Pearson | −0.145 | 0.673 | 0.877 | −0.116 | 0.692 | 0.81 |
| 1995 | 0.485 | Pearson | −0.091 | 0.797 | 0.877 | −0.147 | 0.615 | 0.81 |
| 1993 | 0.763 | Pearson | −0.155 | 0.654 | 0.877 | −0.147 | 0.617 | 0.81 |
| 1991 | 0.295 | Pearson | −0.145 | 0.673 | 0.877 | −0.147 | 0.615 | 0.81 |

correlations are consistently weak (Supplemental Information 9). This lack of statistical strength occurs for subsampled diversity estimates at the two-year intervals for each of ornithischians (Table 1), sauropodomorphs (Table 2), theropods (Table 3), and dinosaurs overall (Table 4), meaning that we cannot confidently interpret anything here. The only time the results come close to alpha (0.05) is for the correlation between Ornithischia and sea level during 2007–2013 ($p$ = 0.062–0.084, $\rho$ = 0.481–0.516), but our correction methods reduce the strength of all our statistical results.

**Table 3 Theropod correlation tests results.**

| Theropods | | | Sea level | | | Palaeotemperature | | |
|---|---|---|---|---|---|---|---|---|
| | Shapiro–Wilk (p) | Correlation test | cor | p | Adjusted p | cor | p | Adjusted p |
| 2015 | 0.036 | Spearman | 0.175 | 0.588 | 0.672 | 0.115 | 0.71 | 0.868 |
| 2013 | 0.098 | Pearson | 0.234 | 0.464 | 0.464 | 0.334 | 0.264 | 0.362 |
| 2011 | 0.027 | Spearman | 0.099 | 0.751 | 0.751 | 0.059 | 0.844 | 0.868 |
| 2009 | 0.032 | Spearman | 0.17 | 0.579 | 0.672 | 0.055 | 0.856 | 0.868 |
| 2007 | 0.029 | Spearman | 0.17 | 0.579 | 0.672 | 0.055 | 0.856 | 0.868 |
| 2005 | 0.072 | Pearson | 0.289 | 0.316 | 0.464 | 0.363 | 0.184 | 0.362 |
| 2003 | 0.027 | Spearman | 0.407 | 0.151 | 0.659 | −0.061 | 0.832 | 0.868 |
| 2001 | 0.006 | Spearman | 0.346 | 0.247 | 0.659 | −0.086 | 0.773 | 0.868 |
| 1999 | 0.028 | Spearman | 0.379 | 0.202 | 0.659 | −0.051 | 0.868 | 0.868 |
| 1997 | 0.193 | Pearson | 0.476 | 0.1 | 0.25 | 0.254 | 0.362 | 0.362 |
| 1995 | 0.107 | Pearson | 0.511 | 0.074 | 0.25 | 0.257 | 0.355 | 0.362 |
| 1993 | 0.101 | Pearson | 0.251 | 0.409 | 0.464 | 0.264 | 0.342 | 0.362 |
| 1991 | 0.013 | Spearman | 0.209 | 0.494 | 0.672 | −0.071 | 0.803 | 0.868 |

**Table 4 Total dinosaur correlation tests results.**

| All dinosaurs | | | Sea level | | | Palaeotemperature | | |
|---|---|---|---|---|---|---|---|---|
| | Shapiro–Wilk (p) | Correlation test | cor | p | Adjusted p | cor | p | Adjusted p |
| 2015 | 0.327 | Pearson | 0.189 | 0.467 | 0.467 | −0.051 | 0.832 | 0.984 |
| 2013 | 0.233 | Pearson | 0.226 | 0.385 | 0.467 | −0.099 | 0.678 | 0.984 |
| 2011 | 0.059 | Pearson | 0.324 | 0.204 | 0.467 | 0.108 | 0.652 | 0.984 |
| 2009 | 0.021 | Spearman | 0.284 | 0.268 | 0.367 | −0.072 | 0.763 | 0.876 |
| 2007 | 0.489 | Pearson | 0.233 | 0.367 | 0.467 | 0.01 | 0.966 | 0.984 |
| 2005 | 0.045 | Spearman | 0.207 | 0.407 | 0.367 | −0.095 | 0.682 | 0.876 |
| 2003 | 0.053 | Pearson | 0.305 | 0.218 | 0.467 | 0.025 | 0.914 | 0.984 |
| 2001 | 0.043 | Spearman | 0.232 | 0.367 | 0.367 | −0.089 | 0.71 | 0.876 |
| 1999 | 0.066 | Pearson | 0.342 | 0.179 | 0.467 | 0.005 | 0.984 | 0.984 |
| 1997 | 0.27 | Pearson | 0.358 | 0.159 | 0.467 | −0.048 | 0.84 | 0.984 |
| 1995 | 0.13 | Pearson | 0.275 | 0.303 | 0.467 | 0.021 | 0.931 | 0.984 |
| 1993 | 0.119 | Pearson | 0.221 | 0.429 | 0.467 | 0.046 | 0.856 | 0.984 |
| 1991 | 0.049 | Spearman | 0.261 | 0.347 | 0.367 | −0.04 | 0.876 | 0.876 |

## DISCUSSION

### The influence of sampling and publication history on dinosaur diversity estimates

The impact of publication history on estimates of both raw and subsampled dinosaur diversity has direct consequences for our interpretation of their evolutionary history and diversification (*Benton, 2008a*; *Tarver, Donoghue & Benton, 2011*). Using a small window of historical discovery, we show that dinosaur diversity remains highly volatile in specific
geographical regions and geological time, typically where sampling levels remain very uneven or the overall sampling pool is very small (*Sepkoski, 1993*; *Alroy, 2000b*). In poorly sampled areas, it is clear that even small changes to the data can yield substantial changes, as we are often dealing with very small total sample sizes. This is reflected much less on an apparent 'global' scale, and much more so when we look at regional signals after subsampling is applied. As the rate of dinosaur discovery is increasing (both taxonomically and for occurrences) (Figs. 1 and 2), we expect this volatility to be present in the future.

As research on dinosaurs continues and new taxa are described and published from existing fossiliferous formations, raw diversity is expected to become more correlated with rock availability as result of increasing sampling effort (*Raup, 1977*; *Wang & Dodson, 2006*; *Benton, 2015*), and represents a form of publication bias (*Sepkoski, 1993*; *Alroy, 2000b*; *Jouve et al., 2017*). Research has shown that new dinosaur discoveries, and changes in their taxonomy and phylogenetic relationships, can strongly influence our understanding and interpretation of their fossil record and diversification patterns (*Weishampel, 1996*; *Tarver, Donoghue & Benton, 2011*). In this study, we examined the historical trajectory of different dinosaur diversity estimates to observe whether sampling curves are beginning to stabilise or not. For raw diversity estimates, we find evidence for relatively stable patterns in spite of any 'bonanza effect' (i.e. fossil discoveries driving formation counts, especially prevalent in Lagerstätten) (*Raup, 1977*; *Benton, 2015*). The fact that the curves remain relatively consistent, despite the variable addition of new taxa, suggests we are seeing some form of the 'redundancy' hypothesis at play, in that fossils and sampling are non-independent from each other, when only raw data are considered (*Benton et al., 2011*, *2013*; *Dunhill, Hannisdal & Benton, 2014*; *Benton, 2015*). Conversely, a more appropriate interpretation might be that we are generally sampling fairly, or consistently, from an underlying occurrence pool through historical time, or that our application of subsampling based on a standardised estimate of coverage is sufficient to eliminate any such sampling biases.

However, what is the explanation for the diversity patterns we obtained so far, and what does the variation in these patterns tell us? Generally, a dinosaur bearing formation availability effect makes the Kimmeridgian, Barremian, Albian, Aptian, Campanian, and Maastrichtian the most productive stages (*Barrett, McGowan & Page, 2009*; *Butler et al., 2011*; *Upchurch et al., 2011*; *Tennant, Mannion & Upchurch, 2016b*). By counting genus density (number of genera per million year), three stages from these stand out: Kimmeridgian, Campanian, and Maastrichtian (*Taylor, 2006*), with Asia being the most productive continent followed closely by North America, then Europe, South America, Africa, Australasia, and finally Antarctica. However, what is clear from our analyses is that this is not historically consistent, and prone to change as new regions are opened up for exploration and discovery.

There is a well-recognised relationship between the amount of rock available for palaeontologists to search for dinosaur fossils, and how this influences our interpretations of their diversity patterns (*Barrett, McGowan & Page, 2009*; *Butler et al., 2011*; *Mannion et al., 2011*; *Upchurch et al., 2011*). This raises questions about the extent to which many aspects of diversity curves could be artefacts caused by changes in global sea levels, tectonics, and other geological processes related to preservational or geological

megabiases (*Peters & Foote, 2001*; *Smith, Gale & Monks, 2001*; *Smith & McGowan, 2007*; *Peters & Heim, 2010*; *Heim & Peters, 2011*; *Peters & Heim, 2011a*; *Smith, Lloyd & McGowan, 2012*; *Smith & Benson, 2013*). As a way of exploring this, *Barrett, McGowan & Page (2009)* applied the 'residuals' method (formerly designed by *Smith & McGowan (2007)* for marine fossil taxa) to account for these sorts of geological biases, and demonstrated that many features of dinosaur diversity curves are sampling artefacts that reflect changes in the amount of fossiliferous rocks and thus reflect geological rather than biological signals. However, this method has received substantial criticism since, and might not be appropriate for studies of palaeodiversity (*Brocklehurst, 2015*; *Sakamoto, Venditti & Benton, 2017*). However, the influence of these geological biases appears to have been largely mitigated in recent studies by considering a historically accurate account of sampling and modelling variation through time (*Alroy, 2010a*, *2010b*, *2010c*; *Newham et al., 2014*; *Mannion et al., 2015*; *Nicholson et al., 2015*; *Grossnickle & Newham, 2016*; *Tennant, Mannion & Upchurch, 2016b*). Here, sampling heterogeneity in terms of both collection effort and rock availability can be accounted for through subsampling methods, which appear to capture and alleviate at least part of the geological signal. These relative changes in the amount of rock available for sampling, the number and abundance of different taxa, and the historical sampling intensity of different rock formations have implications for the patterns of palaeobiological change that we infer from them. An interesting extension of the present study, which explores historical publication bias, would be to test how the historical context of sampling (e.g. outcrop area variation or availability through time, sampling intensity through time) corresponds to our historical estimates of diversity.

We find that there are four main time periods when great caution should be applied to interpreting processes or patterns based on dinosaur diversity, based on volatility in subsampled diversity estimates and coverage levels. These are: (1) the Late Jurassic interval for theropods in Europe, North America, and Asia (Figs. 9 and 10); (2) the mid-Late Cretaceous interval for theropods in South America and Asia (Figs. 9 and 10); and (3) the mid-Late Cretaceous interval for ornithischians in North and South America and Asia (Figs. 6 and 7); (4) the mid-Late Cretaceous for sauropodomorphs in Africa, Asia, and South America (i.e. Gondwana) (Figs. 12 and 13). As well as this, the Late Triassic dinosaurian record is in a state of flux at the present (*Baron, Norman & Barrett, 2017*), and should be interpreted carefully (Figs. 6, 9 and 12). These represent the times when diversity estimates are changing most rapidly due to a combination of taxonomic revision and discovery-driven publication. While we cannot predict the future of dinosaur discovery, or the selective nature of publication, it seems prudent to suggest that we are cautious in our interpretation of events in dinosaur macroevolution in these intervals, similar to the conclusions reached by *Tarver, Donoghue & Benton (2011)*.

## Discovery influences regional patterns of dinosaur diversity through time

### Ornithischians

The J/K interval decline in subsampled diversity remains constant and recognisable throughout publication history, with this stability suggesting a real biological signal and

not a publication artefact (*Tennant, Mannion & Upchurch, 2016b*). However, more focussed sampling needs to occur on J/K interval deposits to reveal the true global signal, as much of this pattern is based on fossils exclusively from historically well-sampled European localities (*Tennant et al., 2016*) (Figs. 6, 10 and 12). Ornithischian subsampled diversity decreases steadily through the Early Cretaceous in Europe, with a possible radiation in the Campanian to Maastrichtian, perhaps explained by an increase of recent occurrences of latest Cretaceous dinosaurian findings mainly in Spain, Portugal, France, and Romania (*Riera et al., 2009*; *Csiki et al., 2010*). However, many of these latest Cretaceous European dinosaur faunas are not particularly well-resolved stratigraphically compared to the well-studied North-American sections, which makes the timing of any regional extinction here and comparison with North America and Asia difficult at the present. Advanced ornithischian faunas, including ceratopsians and hadrosaurids, appear to have diversified extremely rapidly in the latest Cretaceous, but this is classically explained by the oversampling of North American Late Campanian localities, like Dinosaur Park Formation and its approximate temporal equivalents. Although a small rise in subsampled diversity is recovered from the Campanian to the Maastrichtian in Europe, this is considerably less marked than the decline in North America, where subsampling reveals that ornithischian diversity was actually declining from the Campanian to Maastrichtian (*Brusatte et al., 2015*).

Ornithischian subsampled diversity in Asia has been increasing steadily through publication time in the 'middle' Cretaceous, filling in the gap from equivalent latitude European deposits at this time. This is plausibly due to the radiation of Parksosauridae and Ankylopollexia clades, two of the most dominant Late Cretaceous dinosaurian taxa around this time. Together with the North American record, this manifests as a great global decline across the Early–Late Cretaceous interval, a pattern that was not recognised until more recent years due to the discovery of more Konzentrat-Lagerstätten in Mongolia and China around this time, such as the Jehol Biota (*Lambert et al., 2001*; *Godefroit et al., 2008*; *Upchurch et al., 2011*). A perceived Late Cretaceous subsampled diversity increase for Asian taxa, particularly hadrosauroids, could be due to a renaissance in the discovery of Cretaceous Asian dinosaurs over the past two decades (*Lloyd et al., 2008*; *Barrett, McGowan & Page, 2009*; *Zhou & Wang, 2010*; *Upchurch et al., 2011*; *Mo et al., 2016*). Despite the increasing availability of Early Cretaceous dinosaur-bearing formations (DBFs) in Africa in the last 20 years (e.g. Tunisia, Niger; *Taquet & Russell, 1999*; *Anderson et al. (2007)*), sampling here is still too limited to reveal any consistent patterns in ornithischian subsampled diversity (*Mannion et al., 2011*; *Upchurch et al., 2011*; *Tennant, Mannion & Upchurch, 2016b*) (Figs. 6 and 7).

This regional distinction could be due to the tie between ecomorphological function and biological diversity, as Asian hadrosauroids increased in morphological disparity during the latest Cretaceous, whereas in North America large-bodied bulk-feeding ornithischians decreased in their disparity (*Vavrek & Larsson, 2010*; *Campione & Evans, 2011*; *Brusatte et al., 2012*; *Mitchell, Roopnarine & Angielczyk, 2012*). In North America, several abiotic factors, including extreme fluctuations of the Western Interior Sea, and the Laramide orogeny and proposed biogeographic provincialism, may have affected the
evolution of North America dinosaurs in distinct ways from species on other continents (*Gates, Prieto-Márquez & Zanno, 2012*; *Arbour, Zanno & Gates, 2016*), meaning that the North American record is unlikely to be representative of global diversity pattern (*Sampson et al., 2010*; *Brusatte et al., 2012*).

### Theropods

As already shown elsewhere (*Barrett, McGowan & Page, 2009*; *Brusatte et al., 2012*), 'global' theropod diversity trends are overall very similar to that of Ornithischia, with subsampled diversity increases during the Late Jurassic (Oxfordian and Tithonian peaks punctuated by a Kimmeridgian decline), late Early Cretaceous (Aptian), early Late Cretaceous (Cenomanian) and latest Cretaceous. Moderately high Middle and Late Jurassic diversity subsampled levels represent the radiation of major avetheropodan clades, and a wealth of new discoveries in recent years, particularly from Asia (*Upchurch et al., 2011*; *Xu et al., 2011*; *Carrano, Benson & Sampson, 2012*; *Benson et al., 2014*; *Tennant, Mannion & Upchurch, 2016b*).

European subsampled theropod diversity is more constant than in other regions, with a Bajocian peak followed by a Bathonian–Oxfordian trough, and a Kimmeridgian peak followed by a Tithonian to Valanginian drop. This can, at least in part, be explained by an abundance of well-sampled Late Jurassic formations from across Western Europe (*Upchurch et al., 2011*; *Benson et al., 2013*; *Tennant et al., 2016*). Barremian diversity is increasing rapidly through publication history, and is now as high as calculated for the Kimmeridgian. As with the Late Jurassic, at least part of this signal represents the influence of a Lagerstätten effect (e.g. Las Hoyas, Spain) (*Buscalioni et al., 2008*; *Upchurch et al., 2011*; *Sánchez-Hernández & Benton, 2012*), highlighting that single, well-sampled formations can have a profound historical effect on our understanding of regional diversity patterns, even when subsampling methods are applied. The European Aptian–Albian record is increasing slower through time compared to the Campanian–Maastrichtian. However, this might possibly change in the future, as the ichnological record in southern Europe is quite abundant for the Aptian–Albian interval, and suggests a currently unrecognised dinosaurian diversity present there (*Dalla Vecchia, 2002*; *Meyer & Thuring, 2003*).

The North American theropod record is dwarfed by an oversampling of latest Cretaceous dinosaur-bearing formations (e.g. Dinosaur Provincial Park, Hell Creek Formation). An increasingly even representation of latitudinally diverse localities from the Cenomanian–Campanian of Utah, Colorado, New Mexico, and Mexico (e.g. Wahweap Formation), may increase the magnitude of the small subsampled diversity drop through the Maastrichtian. Subsampling highlights a latest Jurassic peak in diversity (due to the abundance of remains from the well-sampled Morrison Formation; *Foster (2003)*), although Jurassic subsampled diversity never attains that of the Cretaceous highs during the Aptian and Campanian. In contrast to *Brusatte et al. (2015)*, who found no evidence for a progressive Campanian-Maastrichtian decline in North American theropod faunas using similar SQS analyses (implemented in R; see (*Tennant, Mannion & Upchurch, 2016a*, *2016b*) and (*Alroy, 2010a*, *2010c*) for comparative discussions), we find a very slight

decline that remains constant through publication history, that likely relates to our usage of a slightly different subsampling approach. Aptian subsampled diversity is relatively high due to the more heavily sampled localities from Montana to Texas (*Kirkland et al., 1997*; *Cifelli et al., 1999*; *Kirkland & Madsen, 2007*).

In Africa, there is a Cenomanian radiation (Fig. 9C) mainly due to the multitaxic theropod dominated Kem Kem beds and other Albian–Cenomanian ('middle' Cretaceous) equivalents in Northern Africa, but this signal might have been altered by time averaging effects constraining a more temporally diluted diversity in a single unit (*Mannion & Barrett, 2013*; *Evers et al., 2015*; *Chiarenza & Cau, 2016*). Asian subsampled diversity peaks in the Aptian, Campanian, and Maastrichtian might be explained by a Lagerstätten 'bonanza' effect, especially considering the high quality preservation deposits discovered and heavily sampled in the last 20 years (e.g. Liaoning) (*Lloyd et al., 2008*; *Zhou & Wang, 2010*; *Godefroit et al., 2013*; *O'Connor & Zhou, 2015*; *Tennant et al., 2016*), although coverage remains only moderate (around 0.5) in each of these intervals (Fig. 10). Similarly to the pattern in Africa and Asia, South American theropod subsampled diversity stands out compared to other North America and Europe, remaining relatively signal deficient. Despite an increasing rate of discovery of new taxa, which often alter our knowledge of dinosaur phylogeny and biogeography from the 'middle' Cretaceous of Patagonia and Brazil (*Novas et al., 2005a*, *2013*; *Novas & Pol, 2005*; *Canale et al., 2009*), coverage remains poor at the stage level, emphasising the need for greater stratigraphic resolution of the theropod-bearing formations here.

### Sauropodomorphs

Subsampled diversity patterns of sauropodomorphs share some characteristics of those of theropods and ornithischians, despite having a different fossil record due to taphonomic differences (i.e. larger, more robust skeletons being preferentially preserved in different environmental settings) (*Mannion & Upchurch, 2010*, *2011*; *Dean, Mannion & Butler, 2016*). This is compounded by a difficulty in assigning a large number of taxa to specific stage bins, which unfortunately excludes many of them from our analyses (Supplemental Information 1). Differences in diversity patterns between sauropodomorphs and ornithischians have classically been interpreted as being due to exclusive competition between the two main herbivorous dinosaurian subtaxa (*Butler et al., 2009*), with an explosive radiation in ornithischians during the Early Cretaceous resulting from the apparent decline in diversity of sauropodomorphs. In fact, the J/K transition represents a major extinction 'event' for sauropodomorphs, reflecting the decline of non-neosauropods, diplodocoids and basal macronarians (*Mannion et al., 2013*; *Tennant, Mannion & Upchurch, 2016b*). Sauropodomorph faunas have a low subsampled diversity in the earliest Cretaceous, coupled with a generally poor fossil record (*Mannion & Upchurch, 2010*), but at a time when we otherwise see rapid increases in theropod and ornithischian diversity and a prolonged phase of faunal turnover (*Upchurch & Mannion, 2012*; *Tennant, Mannion & Upchurch, 2016b*). Sauropodomorph subsampled diversity levels fluctuate from the 'middle' Cretaceous until the final latest Cretaceous radiation, with a possible small decline in the Maastrichtian. This finding is somewhat

contrary to that of *Sakamoto, Benton & Venditti (2016)* who found that their decline was initiated in the Early Cretaceous, and that the diversification of titanosaurs was at an insufficient rate to compensate for the overall loss of sauropodomorph lineages throughout the rest of the Cretaceous. This discrepancy could be due to the differences in datasets used, and that several recently named titanosaurs taxa have yet to be included in published phylogenies, or the distinction between estimation diversity levels against diversification rates. However, we find that sauropodomorphs are at their most diverse during the Albian (Fig. 12). Sauropodomorphs appear to be overrepresented with respect to what we might expect for almost the entire duration of the Jurassic, whereas the opposite is true for the Cretaceous (*Mannion et al., 2011*; *Upchurch et al., 2011*; *Tennant, Mannion & Upchurch, 2016b*). The general patterns of 'global' subsampled diversity shows a steady increase from Middle to the end of Jurassic with a decline through J/K transition (*Upchurch & Mannion, 2012*; *Tennant, Mannion & Upchurch, 2016b*). The relatively high Late Cretaceous subsampled diversity levels can at least be partially explained by the constant discovery of new titanosaurian taxa, especially from Gondwanan continents (*Vieira et al., 2014*; *de Jesus Faria et al., 2015*; *Bandeira et al., 2016*; *Poropat et al., 2016*), and only recently a more appreciated diversity of diplodocoids (e.g. dicraeosaurids, rebbachisaurids) from relatively poorly sampled regions such as Africa (*Mannion & Barrett, 2013*; *Wilson & Allain, 2015*; *Ibrahim et al., 2016*).

Large-bodied sauropodomorph diversity in the Tithonian is certainly influenced by the intense sampling history of the North American Morrison Formation, where there is an unusually high diversity and cranial disparity of megaherbivores within a relatively resource-poor environment (*Button, Rayfield & Barrett, 2014*). Here, high diversity remains in spite of our accounting for large collection biases associated with Konzentrat-Lagerstätten (*Alroy, 2010a*, *2010c*), implying that sauropodomorphs reached their zenith in diversity during the Late Jurassic. Sauropodomorphs appear to be better sampled than theropods and ornithischians in Africa (Fig. 13C), although their records remain largely too inconsistent and patchy record to reveal any major patterns. Asian subsampled diversity is constantly low until the Maastrichtian, where it increases moderately due to a series of recent discoveries from Pakistan and China (*Malkani, 2010*; *Junchang et al., 2013*). However, the Asian Cretaceous sauropodomorph record is otherwise very poorly sampled, especially compared to ornithischians and theropods. This phenomenon could be explained by a taphonomic size bias discriminating against the preservation of larger-bodied animals in pre-Late Cretaceous Konservat-Lagerstätten, while they are more present although more rare in the dense bone assemblages from the latest Cretaceous of Mongolia, China, and India (*Kidwell, 2001*).

There is a notable subsampled diversity decline in European sauropodomorphs through the J/K transition, as with other dinosaurian groups (*Upchurch & Mannion, 2012*; *Tennant, Mannion & Upchurch, 2016b*). This is distinct from results obtained with other methods (e.g. TRiPS, True Richness estimated using a Poisson Sampling) which do not find any evidence for such a decline (*Starrfelt & Liow, 2016*). Subsampling also reveals that sauropodomorph diversity in the latest Cretaceous of Europe was relatively flat. The sauropodomorph record in South America is poor and mostly confined to the Late
Cretaceous, with diversity levels rising and resolution improving through publication time as coverage increases and as new taxa get identified from emerging Patagonian and Brazilian deposits (*Novas et al., 2005b*; *Novas, 2009*).

Here, it is worth noting the distinction between global and regional sauropodomorph records. On a global level, our results provide strong evidence for a substantial sauropod subsampled diversity decline from the Campanian to Maastrichtian. However, this decline is not represented in any of the regional sauropodomorph diversity signals. Instead, the 'global' signal in the Maastrichtian is comprised of a medley of regional records, which are only continuous with the Campanian record in Europe and North America. Therefore, the 'global extinction' of sauropods in the latest Cretaceous is actually due to regionally heterogeneous sampling signals that are summed into a misleading 'global' curve. A similar case can be made for the apparently 'global' radiation in the Albian, which is primarily a reflection of a well-sampled North American Albian sauropodomorph record (Fig. 12). Thus, when looking at diversity signals, interpretation of global patterns without considering structural changes on a regional level is not recommended.

## Limitations of the present study

As we have shown, the interpretation of subsampled diversity estimates in dinosaurs is often highly sensitive to changes in the taxon-abundance curve, and we can further distort this by relying on a historically biased source of data for our analyses. Our overwhelmingly weak correlation results mean that in no cases could we confidently reject any null hypotheses. As such, it is difficult to interpret how the correlations have potentially changed through time. Some of the reasons for this might be that the tests we used are inadequate for picking apart temporal trends over such a long time period, or a small sample size, often with a lot of missing data. Alternatively, it suggests that sea level is a poor predictor of dinosaur diversity at the stage level, and that dinosaur diversity and sea level are perhaps only related on broader temporal scales (*Haubold, 1990*; *Butler et al., 2011*; *Tennant, Mannion & Upchurch, 2016b*). We also only elected to use a single autocorrelation model, and it would be interesting in the future to explore modelling a wider range of serial correlation structures on palaeontological data, and the impact this might have on correlation analyses. Alternatively, our choice of using genus-level data might have been influential (see *Benton, 2008a*; *Benson et al., 2016*), despite previous assertions that the species and genus level diversity curves for dinosaurs are quite similar (*Barrett, McGowan & Page, 2009*). Future research could investigate the influence that taxonomic resolution has on our interpretation of dinosaur evolution, as well as the influence of changing taxonomic opinions through time (Fig. 3). In addition, as mentioned above, it might simply be inappropriate to analyse 'global' correlations between diversity and extrinsic parameters, due to the regionally heterogeneous nature of diversity data. However, what we do see is that the strength of the relationship between sea level and subsampled diversity, despite being consistently weakly statistically supported, is contingent on the publication history of the group. This lends some support to the recent analysis of *Jouve et al. (2017)*, who also found that small changes in the taxonomic

composition of a dataset can lead to divergent interpretations of the environmental regulators of diversity, although this phenomenon requires further investigation.

The accuracy of the results from the Gondwanan continents should be treated with caution, as it is clear that the fossil record is substantially patchier than the Laurasian record, reflected in the publication histories of specimens from these regions. High-magnitude changes in even moderately well-sampled intervals through publication history suggests we should acknowledge the limitations of any biological interpretations of the dinosaur record in Africa and South America until more reliable data are obtained (*Barrett, McGowan & Page, 2009*; *Mannion et al., 2011*; *Upchurch et al., 2011*; *Tennant, Mannion & Upchurch, 2016b*).

We did not test for how changes in the stratigraphy of dinosaur-bearing formations through time (e.g. as chronological dates are found or refined) influences the structure of sampling pools in each time bin, a factor which is under-studied in palaeodiversity reconstructions (*Gibert & Escarguel, 2017*). Furthermore, by explicitly excluding occurrences that did not fit within a single stage-level time bin, we influence what data are not included in our analyses by rejecting specific formation pools from bins. This will have a particularly stronger effect in formations that span multiple time bins, as well as in formations that have less well-studied chronostratigraphy or less accurate dates. Furthermore, we used stage-level bins that are inherently of uneven duration, as opposed to other commonly used methods such as 2/9/10/million year approximately equal duration bins (*Wang & Dodson, 2006*; *Barrett, McGowan & Page, 2009*; *Butler et al., 2011*; *Upchurch et al., 2011*; *Brusatte et al., 2012*; *Lloyd, 2012*; *Mannion et al., 2012*); there is currently little consensus on which time binning methods are most appropriate for the fossil record, although different bins can influence resulting diversity estimates (*Tennant, Mannion & Upchurch, 2016a*).

The impact that all of these factors can have on diversity estimates is an ongoing discussion in research about palaeodiversity, and exploring them all is beyond the scope of the present study. What is more important for us in terms of study design was the focus on understanding the impact of a single factor that could be compared through publication history, which is what we performed. That is not to say that each of these factors do not also variably influence diversity estimates through time, and investigating how these potential stratigraphic biases influences diversity estimates would be a useful future research avenue.

## CONCLUSION

In this study, we investigated diversity trends through time for three major clades of Dinosauria (Ornithischia, Sauropodomorpha, and Theropoda), by reducing a primary dataset of body fossil occurrences by progressively removing publications at each two-year intervals, up until 1991. By analysing both empirical and subsampled curves, we have been able to see how publication history influences different estimates of dinosaur diversity.

Subsampling reveals that there are major discrepancies between the 1991 and 2015 curves for theropods in the Oxfordian, Aptian, and Cenomanian, for ornithischians in the late Early Cretaceous, and for sauropods in the Albian and latest Cretaceous.

However, almost without exception, these seemingly continuous 'global' diversity patterns are the product of summing together different, and invariably patchier, continental signals with vastly different trends, reflective of distinct geographic sampling histories. In ornithischians, a J/K transition decline is based almost exclusively on European fossils, and a perceived global reduction in their diversity in the latest Cretaceous is the result of an overpowering North American signal. Similarly, 'global' subsampled theropod diversity is prevalently based on the European record, with Asia and North America contributing substantially more after the earliest Cretaceous hiatus. Theropod diversity in the latest Cretaceous is changing the most rapidly compared to any other time interval. In these places where see the most volatility in both subsampled diversity and coverage, we should be careful not to over-interpret patterns, especially in the context of apparent radiations and extinctions. Gondwanan dinosaurian faunas are still relatively poorly sampled despite intensive exploration in the last 20 years, and we expect the influence of discovery in Africa and South America to become more important in the future. Based on this, we urge caution in any evolutionary interpretations relying on Gondwanan dinosaur diversity until sampling improves.

However, the results of this study should be of interest to those who use occurrence-based compilations like the Paleobiology Database that rely heavily on the published literature, especially when ongoing research can potentially dramatically alter our understanding of the evolutionary history of dinosaurs (*Baron, Norman & Barrett, 2017*). Both the addition of new taxa, and new occurrences of existing taxa, are clearly important in establishing stable and re-usable diversity curves for further research, and the maturity and growth of taxonomic datasets must be assessed prior to further macroevolutionary study (*Tarver, Donoghue & Benton, 2011*). By neglecting the publication history, and potential biases involved in this, we open ourselves up to potentially misinterpreting the patterns and processes involved in dinosaur evolution. In light of this, it is possible that many previous dinosaur diversity studies are likely now incorrect due to the large number of new discoveries being made every year (Figs. 1 and 2). Furthermore, it is also likely that the analyses presented in this paper will be demonstrated to be wrong in several years' time, and it remains to be seen whether we will be ever able to faithfully reconstruct an accurate diversity curve for Dinosauria.

Future research could investigate the impact that variation in taxonomy, systematics, and validity of dinosaur taxa through publication history (*Benton, 2008a*, *2008b*) (Fig. 3) has on diversity, and the influence that changes in the historical quality and stratigraphic resolution of the fossil record has on this. Furthermore, given the importance of sampling biases on our interpretations of the dinosaur fossil record (*Barrett, McGowan & Page, 2009*; *Butler et al., 2011*; *Mannion & Upchurch, 2011*; *Upchurch et al., 2011*; *Benton, 2015*; *Tennant, Mannion & Upchurch, 2016b*), research could look at how the relationships between sampling proxies and dinosaur diversity change through historical time.

## ACKNOWLEDGEMENTS

We are grateful for the combined efforts of all those who have collected Triassic–Cretaceous dinosaur data, and to those who have entered these data into the Paleobiology

Database, especially J. Alroy, M. T. Carrano, P. D. Mannion, R. B. J. Benson, and R. J. Butler. We also thank J. Alroy for providing the Perl script used to perform SQS analyses. This is Paleobiology Database official publication number 304. We would like to extend our thanks to David Button and the four other anonymous referees for their detailed and constructive reviews, as well as comments from Manabu Sakamoto, which greatly helped to improve this manuscript. We would all like to thank Serjoscha Evers for helping to come up with this idea during Progressive Palaeontology 2016 and during subsequent discussions.

### Funding

Jonathan P. Tennant was funded by a National Environmental Research Council PhD studentship. Alfio A. Chiarenza is funded by an Imperial College London Janet Watson Departmental PhD Scholarship. Funding for Matthew Baron was provided by a NERC/CASE Doctoral Studentship (NE/L501578/1). The funders had no role in study design, data collection and analysis, decision to publish, or preparation of the manuscript.

### Grant Disclosures

The following grant information was disclosed by the authors:
National Environmental Research Council.
Imperial College London Janet Watson.
NERC/CASE Doctoral Studentship: NE/L501578/1.

### Competing Interests

The authors declare that they have no competing interests.

### Author Contributions

- Jonathan P. Tennant conceived and designed the experiments, performed the experiments, analyzed the data, prepared figures and/or tables, authored or reviewed drafts of the paper, approved the final draft.
- Alfio Alessandro Chiarenza conceived and designed the experiments, authored or reviewed drafts of the paper, approved the final draft.
- Matthew Baron conceived and designed the experiments, authored or reviewed drafts of the paper, approved the final draft.

### Data Availability

The Open Science Framework: https://osf.io/nuhqx/
Identifiers: DOI 10.17605/OSF.IO/NUHQX | ARK c7605/osf.io/nuhqx

### Supplemental Information

Supplemental information for this article can be found online at http://dx.doi.org/10.7717/peerj.4417#supplemental-information.

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
