# Peer review of "How has our knowledge of dinosaur diversity through geologic time changed through research history?"

_PeerJ, doi:10.7717/peerj.4417_

## Round 0.1 · original submission · Major Revisions

Dear authors,

I have received three detailed reviews for your manuscript, these are appended below. You will see that the reviewers have made quite a few suggestions that need to be addressed. Particularly, the reviewers have expressed some concerns over the purpose of the study and the conclusions that may be drawn from the results in their present form. In part, some of these concerns may have arisen due to the omission of some details in the methods and the difficultly the reviewers experienced in interpreting the figures as illustrated. All reviewers have also noted that aspects of the language could be improved. Please avoid using vague wording, this will reduce the accessibility of your paper and cause confusion for the reader.

In general, the reviewers are complimentary about the effort that the authors have put into this study, and I agree - I would be happy to consider a revised version of your manuscript.

Reviewer 1 ·

Basic reporting

The purpose of this paper could be made clearer. We know, for example, that our knowledge of the fossil record can change over time, but the figures all seem to show generally the same patterns of rises and falls. Some recommendations (section beginning L604) are made for future data collection, but the applied approach (SQS) can (and does) already correct for these biases (see Alroy 2010a-c, already cited in the manuscript and the notes in the Perl script included in the supplementary information). Similarly the correlations seem problematic as there is no clear precedent for either sea level or temperature to be correlated with dinosaur diversity (see also comments below). More broadly I'm not sure what to take away from the manuscript. At present there is only a weak attempt to establish predictive power for future dinosaur richness curves (L401-413). However, I am not convinced that the logic holds here (i.e., the largest differences between 1991 and 2015 richness correspond to the time bins where we can expect greatest volatility in future estimates). Instead it would make more sense to ask what drives the largest changes? For example, is it the discovery of new, or increased exploration of older, formations/basins that drives change? This would add the greatest relevance to the data. I also strongly feel that taxonomic revisions should come in here too, but these are not currently included in the data (see below).

Another potential question currently unaddressed, but potentially addressable concerns a different relationship - that between richness and sampling proxies. The authors themselves point out that this would be interesting (L365, L411-413), and this data is easily gleamed from the Paleobiology Database. This has already been partially addressed by Benton (2015, already cited), but there is clear room to improve and expand on his analysis. I would strongly urge the authors to consider addressing this as I think testing the hypothesis that sampling and richness become increasingly decoupled would lead to the most interesting result possible from their data (regardless of outcome).

The language is mostly clear but could be improved by increased brevity/formality (e.g., "increasingly more even", "newsworthiness") and although the reference list is large and the discussion well cited a number of key papers could be mentioned. For example, some other important historical approaches to the dinosaurian record, albeit with more of a phylogenetic focus, are Weishampel (1996) and Tarver et al. (2011). These seem worthy of mention, if minimally to establish precedent. (See also additional references below.) The title should also be changed as it uses time twice in completely different ways but without appropriate context. I suggest delineating these as "geologic" vs "historical" time.

There is currently a complete absence of the model/correlation results from the main manuscript which seems like a major omission. (I had to dig through the supplementary files to find these.) These should be moved to the main text as a table or tables. Otherwise I felt that the discussion focused more on dinosaurian diversity in general rather than on specific changes to the fossil record over the 24-year sampling interval. For example, the Jehol biota has dramatically altered our picture of the origin of birds, and the position of therizinosaurs has been settled upon. More recently, of course, the Baron et al. (2017; Nature) paper has rearranged key clades at the base of the dinosaur tree which can affect our understanding of the completeness of the early part of the Mesozoic dinosaur record.

As a final point the authors discuss the "redundancy hypothesis" (L376-377), but do not defend its criticisms of sampling corrected diversity (i.e., SQS), which weakens the manuscript. This point should be addressed directly.

J. E. Tarver, Donoghue, P. C. J. and Benton, M. J., 2011. Is evolutionary history repeatedly rewritten in light of new fossil discoveries? Proceedings of the Royal Society B, 278, 599-604.

Weishampel, D. B., 1996. Fossils, phylogeny, and discovery: a cladistic study of the history of tree topologies and ghost lineage durations. Journal of Vertebrate Paleontology, 16, 191-197.

Experimental design

In its current form many aspects of the experimental design are not made clear and there appear to be some critical omissions that the authors should minimally address in the text as they potentially strongly bias their results, if not modify their analyses to more directly account for them.

The most critical of these is that their historical approach is likely very misleading as it ignores other changes than just sampling. Specifically, we know that taxonomy, and dinosaur taxonomy in particular, fluctuate dramatically over time (Benton 2008), and can even be heavily influenced by single workers (Benton 2010). Here the authors appear to not correct for this, and thus their 1991 data adheres to 2015 (or likely even more recent) taxonomic opinions. Taxonomic opinions can be extracted from the Paleobiology Database and a new synthesis constructed, so this could be addressed directly. At present using modern taxonomic opinions will lead to the *appearance* of a closer relationship between 1991 and 2015 data that is bias and not signal. In a similar vain the stratigraphic resolution of fossil occurrences tends to become refined over time, as does the geologic time scale. Both these factors are likely to influence the data, although the former is likely to be harder to address and the latter may not be relevant if binning data by geologic stages. However, these should be discussed as we have seen, for example, Chinese formations thought to contain birds older than Archaeopteryx be re-dated as Early Cretaceous.

Some other issues likely just need clarification. For example, the authors use SQS, but key details are missing. One obvious one is the quorum level used. From examination of the Perl script a value of 0.4 seems to have been applied, but no justification for choosing this is made. In fact the authors later suggest multiple sampling levels should be used (L587-588). I think this is worthy of exploration. For example, lower quorums would (potentially) allow more data to be returned, and hence can modify the correlations (see also comments below). Similarly a major issue, already discussed in places by the authors, is that the Good's U term will be biased by the large number of singletons that are inevitably generated from publication-formed databases (L548-549, L610-612, L620-621). Implementations of SQS has always had some form of correction for this, for example the "single publication occurrence" correction (Alroy 2010a-c). However, it is not clear what correction (if any) the authors are using here. This is a critical point as without such a correction the SQS results are compromised.

Some changes to the correlations could also be considered. Currently it appears that the length of time series changes dramatically (based on the figures), i.e., 1991 time series often have missing bins that are filled in the 2015 data. This makes comparison across correlations difficult as any changes could be attributed *either* to the additional data *or* to the absence of key bins from the correlations. There is no clear way to fix this however, except perhaps for only using data from bins present in both 1991 and 2015 for each correlation. Similarly, it would be interesting to show graphically how the correlations change across *every* two-year value from 1991 to 2015. Are the changes monotonic or are there big shifts? If the latter, what changes to the fossil record precipitate these? It would also be interesting to correlate diversity with diversity and not just the external proxies. Visually it seems that there is not a major difference in many cases (i.e., data subsets), but it would be good to quantify these.

Finally, the authors use a genus-level approach, but others have noted how this can confound correlations between richness and non-biogenic time series (e.g., Wiese et al. 2016). As this is a primary aim of the manuscript this point should minimally be addressed in the text if not through quantitative analysis (i.e., a separate set of analyses at the species-level).

Wiese, R., Renaudie, J. and Lazarus, D. B., 2016. Testing the accuracy of genus-level data to predict species diversity in Cenozoic marine diatoms. Geology, G38347.1.

Validity of the findings

There are potential issues with both the SQS approach (singleton correction) and correlation (missing bins) that urge some caution in accepting the results as they stand (see detailed comments above). The database used is generally very solid for dinosaurs, but critical aspects (taxonomic opinion shifts) are not being exploited. The conclusions could be much briefer and stronger (e.g., after addressing for taxonomic changes over historical time).

Additional comments

No further comments.

Reviewer 2 ·

Basic reporting

The article is generally well written but the authors regularly lapse into a journalistic style of writing that you would not associate with the primary scientific language. This makes certain things difficult to understand, especially to anyone who is not a native English speaker. Please stick to unambiguous and professional language throughout. I have highlighted some examples below, but there may be more instances of this throughout the text:
line 101: “secular variation” I’m guessing you mean temporal variation?
line 119: “volant” I’m pretty sure this is a French word…
line 442: “Judithian” I’ve no idea what this means…
line 505: “South America theropod diversity sparkles” what does “sparkle” mean in this context? Glammed up?

Acknowledgements: Whilst the authors are free to include whatever they like in an acknowledgements section, I personally feel it is better to keep things professional and to avoid making the reviewer vomit on the manuscript.

There are parts of the manuscript, mostly in the discussion, that suffer from poor sentence structure and writing style. As they stand it is difficult to extract anything meaningful from these statements. A few examples:
Lines 362-363: This sentence doesn’t make any sense to me.
Lines 384-386: Poor sentence structure.
Lines 390-394: Poor sentence structure, what is a structural bias? To a geologist, structure implies faulting and folding.
Line 517: Divergence is not the correct word to use here as it implies they have moved away from one another – better to simply use difference.
Lines 533-535: Doesn’t make sense.
Lines 535-536: Full of typos and doesn’t make sense.
Lines 536-541: Consider rewriting. It is often confusing when you conflate geological and human timescales in a single sentence. I realise this is essential considering the topic, but be careful to explain yourselves better. Non-geoscientists have enough trouble with geological time as it is.
Lines 549-550: Rephrase to: …”implying that sauropodomorphs reached their true zenith in diversity during the Late Jurassic”
Lines 559-563: Rephrase, possibly split into two sentences.
Lines 592-596: Rephrase, possibly split into two sentences.
Line 605: It is odd to start a paragraph with “Often,”
Line 635: doesn’t make sense – I think you’re missing a “by” and “a” either side of reducing.
Lines 651-653: Poor sentence structure.

Figures:
I find the figures difficult to interpret, even for somebody who has extensive knowledge of the geologic timescale. There is no indication as to what the vertical solid and dashed red lines represent. Are these significant? I guess they are Period and Epoch boundaries. They should be clearly labelled in the figure legends. A biologist looking at these plots will not have a clue what they represent.
Geological periods, epochs and stages need to be clearly labelled on the y axes of all the time series as, in the manuscript the authors refer to these chronostratigraphic divisions rather than figures in millions of years. In that sense, the text and figures are incompatible and it makes checking the results descriptions very time consuming. I’m guessing the figures are produced in R? If so, simply use the geoscale package instead of the standard plot() function.

Raw data is all supplied as supplement.

Experimental design

The research represents original primary research and offers a novel approach to a broad question previously addressed by others which, oddly, the authors do not reference. See (Benton 2008b, a; Tarver et al. 2010).

The research question is clearly outlined and defined. However, I do not feel it is particularly relevant to furthering our understanding of the fossil record or dinosaur evolution. I don’t see how this paper fills a “knowledge gap”.

The analyses appear rigorous and well executed. However, to be certain, I would like much more information about how the subsampling was carried out. SQS is very sensitive to a number of things that the authors do not mention. What quorum levels were used? How many iterations were employed? Were the most common taxa excluded? Were the largest collections excluded? Were singletons excluded? Were adjustments made for taxa occurring in more publications? Any parameters employed need to be clearly outlined in the methods. As it stands, it is impossible to tell if the authors are using appropriate methodology (I think they are) and it is impossible to replicate.

References:
BENTON, M. J. 2008a. Fossil quality and naming dinosaurs. Biology Letters, 4, 729-732.
--- 2008b. How to find a dinosaur, and the role of synonymy in biodiversity studies. Paleobiology, 34, 516-533.
TARVER, J. E., DONOGHUE, P. C. J. and BENTON, M. J. 2010. Is evolutionary history repeatedly rewritten in light of new fossil discoveries? Proc. R. Soc. Lond. B, 278, 599-604.

Validity of the findings

Overall, this manuscript has left me feeling somewhat underwhelmed. To be completely honest, I feel like I have learnt nothing from reading it and I’m wondering what the point of the study actually is. Are the authors wishing to highlight that even though we have found a lot of new dinosaur taxa in the past 15 years or so, it hasn’t really altered our view of relative diversity change but, changes in the sampled abundance of dinosaur fossils do alter our view of dinosaur macroevolution? I find that this conclusion, stated in the final sentence of the abstract, is not supported by the figures, which show very little change in either raw or subsampled diversity trends. Only magnitudes in raw diversity appear to have drastically changed and, given the increased sampling over the past 15 years, this is to be entirely expected. The fact that we have found so many new dinosaurs in the past 15 years HAS increased our knowledge of dinosaur macroevolution. This HAS altered our understanding of dinosaur phylogeny, the evolution of feathers, dinosaur colouration, but name a few. I just don’t see why we should be so preoccupied with working out global richness through time, particularly in a group with such a poor empirical fossil record?

So, my opinion is, that although the authors appear to have been rigorous in their analyses, it don’t feel that this paper adds anything new to the wider argument of the quality of the fossil record or the story of dinosaur diversity through time. I see this sort of study as navel-gazing, something that palaeontology has been criticised for doing for decades by other branches of the life sciences. I realise that the PeerJ publishes manuscripts based on their scientific validity, rather than their significance so, I guess this paper could be published, albeit after a major overhaul to address the many problems stated below, as it might attract some citations from those wishing to find quantifiable data on the sampling history of the fossil record. However, I don’t think it actually adds anything new to our knowledge of the fossil record or dinosaur macroevolution, as the authors repeatedly claim to the contrary.

Perhaps most concerning is that the graphical results (i.e. figures) and results and discussion in the text do not always match up. This may be somewhat because the figures are hard to read in terms of lacking chronostratigraphic labelling. But, in some instances it appears the authors are blatantly misinterpreting the trends in the figures. I do assume that this is done unintentionally but, is nonetheless very concerning.

I don’t wish to be overly negative and I have made some suggestions to how the manuscript can be improved. However, none of these comments really address the wider picture of the relevance of such a study. I feel that it should be more than just a historical account of sampling, perhaps the authors could do a better job of explaining the relevance to me. I feel that if the relevance of the paper is not apparent to me, then it will be lost on the majority of the readership who do not share my interest in the nitty gritty of the quality of the fossil record.

Additional comments

Change the title. I think the authors are trying to be witty and it doesn’t work. It reads more like a mistake in the repeated “through time”. I suggest changing it to something more legible, like “How has our knowledge of dinosaur diversity changed through research history?”

Lines 8-9: Is this really the case? Our current knowledge doesn’t really depend on how this taxonomic archive varies through historical time. We could have found all the dinosaurs yesterday and still have the same picture. All it highlights is the sensitivity of the fossil record to new findings.

Lines 9-12: What are these recent studies that have assessed dinosaur diversity based on raw species? I’m not aware of them.

Lines 12-13. Why does this matter? I’m not saying for certain that it doesn’t, but the authors don’t justify why it matters, they’re just saying it does. I don’t see the logic.

Line 19: Would it be better to replace “overall shape” with “relative trend”?

Lines 24-25: What do you mean by continental signal? This suggests “on land”, when all the data is terrestrial? Do you mean the regional subsets? Be specific. Also, the fact that the subsampled data shows greater changes is quite suspicious to me. Suggests a patchy record that does not have sufficient data for the task.

Lines 27-30: I don’t understand this conclusion. Surely that is obvious? Finding new species changes the pattern and finding new specimens of old species fills in range gaps?

Lines 37-39: I disagree, I think it’s a fool’s errand. Much better to form a specific question and use the appropriate methods to address the incompleteness of the record that are specific to that question.

Line 42: Fossilworks and the Paleobiology Database are essentially the same thing. They are mirrors of one another in terms of the database content. This reads as if they are two different databases which is very misleading for anyone who is not familiar with the politics of the Paleobiology Database (which is pretty much everyone).

Lines 44-46: I don’t think you need this sentence or this figure to promote or justify the use of the PBDB. It is an established resource for this type of analysis.

Lines 47-50: I would widen this discussion to the whole fossil record, not just these databases. The same issues pertain if you are analysing data from a field season.

Lines 47-60: This is a good paragraph to reference the Benton and Tarver et al. papers mentioned earlier.

Lines 61-65: You need to reference these statements if you are claiming people are saying these things. I’m not sure anybody still says the first statement with any certainty.

Lines 69. Again, they’re the same thing.

Lines 75-78: we see some promise here that is then never seized upon again in the discussion. An in-depth analysis of how the taxon-abundance distribution changes through time would, perhaps, be more interesting.

Line 79: what do you mean by a mature data set? It is different to well sampled, no? Make sure you explain this as it might not be obvious to all readers.

Lines 94-97: This suggests we are dealing with a very poor fossil record that is extremely sensitive to minor changes in taxonomy. The same cannot be said for the marine bivalve record, for instance. I think much of this paper is routed in this, we have a very patchy and poor record which is sensitive to the addition of new data.

Line 98: what do you mean by reading and how does it differ from understanding?

Lines 99-105: why bother stating what the paper doesn’t do? Concentrate your energy on explaining what it does do because, to be frank, I’m not that sure I understand that. Also, you have a section on model fitting despite saying you are not going to do this (i.e. point (3)).

Lines 106-107: This is not a hypothesis, it is a question.

Lines 108-109: This needs rephrasing and, is it true? Need a reference to support such a statement.

Lines 114-116: It is not clear where your data comes from. Is it a direct download from PBDB? Or is a hybrid data set derived from Carrano et al. 2015 & Tennant et al. 2016b? Why not use an up-to-date PBDB data set from 2016/17? Why no Triassic data? We have lots of Carnian-Rhaetian dinosaurs – if you’re going to exclude them you need to say why.

Lines 122-125: Can you demonstrate this?

Lines 134-136: I don’t see the logic here, why not include taxa that span multiple stages? Are there many genera that span multiple stages? This sort of approach would destroy most marine invertebrate data sets.

Lines 144-146: Needs explaining better. Do you mean the first occurrence or all occurrences?

Line 149: This is still debateable – i.e. we have no concrete evidence. I’d change it to something like “has been strongly suggested to be”

Lines 154-158: I can’t help but think there is a flaw in this way of thinking. SQS is sensitive to changes in the abundance distribution, right? So, if adding new taxa to a database drastically alters our subsampled richness estimates, then there are questions about whether the database is adequate for this type of analysis in the first place.

Line 160: This shouldn’t be called model fitting as you are not fitting any models! You cannot call correlation tests modelling as modelling means something is being modelled, i.e. predicted. The authors will, no doubt, be familiar with that old saying, “correlation does not equal causation” and therefore we cannot imply that one variable predicts the other when using pairwise correlation statistics. Also, the authors say that they aren’t doing any model fitting earlier in the manuscript.

Lines 164-171: Is this the same as generalised differencing? We’re basically looking at static differencing between adjacent time bins.

Lines 175-177: Why not pick the test whose assumptions best fit the data? I imagine that the Pearson test is violated by the non-normal distributions here as you say later that the results differ between the Pearson and Spearman tests. Drop the Pearson and stick with the Spearman in the name of conservatism and simplicity.

Lines 178-181: the BH correction helps prevent type-I errors not type-II. With multiple comparisons there is an increased chance of a type-I error, i.e. the likelihood of incorrectly rejecting a null hypothesis i.e. a false positive.

Lines 183: I know what they mean, but they need to explain what a family of analyses is.

Lines 185: As well as removing the Triassic, the authors now state they are removing the Early-Mid Jurassic – why?

Lines 188: You don’t use any model fitting approach so you don’t need to mention it at all.

Lines 196-196: Not sure I’d call it consistent. The “shape” (whatever that means) seems to alter dramatically from a roughly steady increase to an exponential Late Cretaceous increase. I find there to be a commonly occurring discrepancy between the results descriptions and interpretations and the figures. This may be exacerbated by the lack of time period labelling on the figures, but it is concerning, nevertheless.

Lines 208-209. Does not make sense, how can something be “stable to changes”? Also, looking at figure 1, ornithischian and theropod patterns are quite different.

Section 217-243: Are we interpreting subsampled diversity as biological diversity? It appears to read that way. However, given the patchiness of the dinosaur fossil record, I feel this is dangerous ground. Would we really expect dinosaur diversity to rise and fall in such an erratic manner? I think the authors need to clarify their standpoint here. Using terms like “diversity crash” seems to infer literal biological interpretation.

Line 295: What is this signal though? Does it actually mean anything when the rest of the record is so poor that you get nothing back?

Lines 298-300: Anything described in the results should also be plotted. We just have to take the authors word for it at this point. This is insufficient.

Lines 308-310: I would not call anything in this plot as strong evidence for anything. OK, so we see a reduction in relative subsampled diversity from one time bin to another. I think the authors need to be cautious of over-interpreting the subsampled results, given the patchy nature of the raw data.

Lines 320 onwards: where are the South American plots? We should be able to see them if you arte describing them.

Lines 330-331: where are these results? If they are important (i.e. mentioned in the paper) they should be summarised in a table.

Lines 334-335: How have the authors got from correlation coefficients to variance explained? Without the correlation results it is difficult to know how they have calculated this. 50% (i.e. R2) explained variance would be a correlation coefficient of around 0.7. Is that how the authors calculated this? I’m also not sure about the logic behind squaring a Spearman rho value as this would just give you the shared variance in the ranked variables. I personally find this to be rather dodgy ground though, as you can’t equate correlation to any sort of predictive model.

Lines 339-341: As I said earlier, I suspect Pearson assumptions are being violated somewhere along the way. I think it would be better to stick to Spearman.

Lines 340-341: Which is? This differs per discipline and in a cross-disciplinary journal like PeerJ you need to be specific. Life sciences = generally < 0.05, right?

Line 358: Needs to be included in the main text as it is essential for independent interpretation of the results. Again, there is no model fitting going on here.

Line 361: You don’t analyse any geological effects.

Line 362: Beginning of sentence doesn’t really make sense. Dinosaur diversity doesn’t really change through publication time.

Line 366-367: I don’t know what a bibliographic form of publishing bias is. Bibliographic just means relating to bibliography?

Lines 367-370: Did you test this? If so, where are the results and conclusions for this? Other than the rather tentative statement below.

Line 409: What do the authors mean by “fickle nature of publication”? When would new fossil finds not be published? I find this a very odd statement indeed.

Line 412: How would you test how outcrop area or availability through time has changed? Do you mean how much outcrop has been sampled up to a point in time? I think that would be nigh on impossible to quantify.

Lines 417-418: Really, where? In the raw and global subsampled data the decline is a minor one which does not stand out against any other changes in diversity through the time series. It is then absent in all of the regional curves, raw and subsampled. I would suggest this is more likely an artefact of summing the regional curves, which the authors highlight as a problem later. Also, you don’t need “either” in this sentence as there is not alternative explanation offered.

Lines 420-421: But there is no J/K boundary decline signal in the European data? This is what I mean when I mention the discrepancy between the figured data and the results and discussion in the text. It is concerning if the authors are misinterpreting their own results in such a blatant fashion.

Line 428: I know what a DBF is, others won’t. I don’t think you define it earlier. Either way, it isn’t used consistently throughout the manuscript so I would avoid using an acronym all together. Just call it what it is, a dinosaur bearing formation.

Lines 451-452: If you’re going to refer to ecologically studies, link it in to your data in a more convincing fashion. The authors use a number of these throwaway references to back up what are at best, tenuous patterns. Personally, I don’t think this reference adds any credence to the argument you a trying to make.

Line 459: be careful about using “significant” when you haven’t tested for significance. It’s trivial, I know, but people get upset about this type of thing.

Line 263: I thought you’d excluded the Triassic? Why are you commenting on a time period you haven’t analysed?

Lines 501-502: Why does this “refined” approach link to this finding? Give me a statistical explanation of this. You can’t just say this and not say why.

Line 503: what does “more-sampled” mean? More localities sampled or localities are more intensely sampled?

Line 515: this is not supported by your data as you don’t have any Triassic-Early Jurassic data, right?

Line 530-531: you criticise the use of DBFs as a proxy earlier in the manuscript, now you adopt it as a benchmark of sampling intensity. Be consistent. Also, don’t use the acronym (as stated earlier).

Lines 543-544: this is another throwaway statement, like the food web one earlier, that neither supports nor refutes your standpoint. So, there is a diverse community caused by niche-partitioning but it is also oversampled? How do you know it is oversampled? What does oversampled mean in this context? Have we found everything?

Line 545: what is the evidence that sauropodomorphs are better sampled? Is it just because there are more fossil occurrences? You need a metric or, at least, some logic to back up statements like these.

Line 574: what is the quorum level? I made this point earlier.

Line 587: What additional statistical analyses?

Lines 592-596: You don’t fit any models so please change this statement. What size is n? No predictions are made so you can’t say that sea level is a poor predictor. I think the final statement sums things up quite well here, the record is so poor that slight changes in taxonomy can upset largescale interpretations.

Lines 605-612: Although there is an element of truth to this across all life sciences, i.e. the publication of new material takes precedence over the fleshing out of data sets, I find some of this section to be irrelevant to the study presented here. After line 607, the attack on the “journals that will publish anything…” comes across as rather petulant and is completely out of context for this study. Surely, for this sort of data set, it does not matter where data is published, so long as it is published somewhere? There is always a vehicle for publication of data sets somewhere. I shouldn’t think a dinosaur occurrence is more likely to enter the PBDB if it is published in Nature compared to one being published in, for example, Proceedings of the Yorkshire Geological Society, a fine journal albeit one with little significance when it comes to “impact”. Therefore, I strongly urge the authors to tone down this section as, presently, it appears to be a side-swipe stemming from a personal agenda rather than being an integral part of this particular study.

Line 621: Explain what you mean by us relying on a biased source of data? Is this a choice we have? Because the way this is written it sounds like you are suggesting that it is.

Lines 624-631: I urge the authors to remove this section. It comes across as very patronising. I would argue that everyone reading this paper will be familiar with the need to collect or record all fossils during their field seasons.

Line 654: Why is global is quotation marks? I think that’s a standard use for the term.

Line 654: Are you certain of this? Where is your proof?

Line 659: You haven’t really tested for extrinsic drivers. Drivers implies causality, you didn’t test for causality.

Line 665: Whose evolution?

Line 666: In think the closing statement is nonsense and closes the paper on a moot point, which is very disappointing.

Reviewer 3 ·

Basic reporting

The manuscript is well structured in a way that is easy to follow despite quite a number of minor grammatical and spelling errors throughout the text. Below, I suggest general and notable comments below that should be addressed by the authors before the manuscript is accepted for publication. More minor and specific comments that should also be considered are embedded in the attached annotated PDF.

1. Vague adjectives are used many times throughout the text, particularly when reporting results without offering precise and clear definitions (e.g., "numerous," "substantial," "massive increase," "moderately strongly," "hyperdiversified"). These should be replaced with specific magnitude, more precise words, or clearly defined. Some, but not all, of these are indicated in the attached annotated PDF. I recommend that any similarly vague words to be replaced or clearly defined as much as possible.

2. The in-text citations are not ordered by publication year (e.g., Ln 72: "Benson et al. 2016a; Benson & Butler 2011; Benson et al. 2016b; Brocklehurst et al. 2017 [...]"). When multiple references are cited within a parenthesis, they should be in chronological order of publication.

3. The order in which results from three dinosaurian clades are reported changes between global patterns of diversity (theropods > sauropodomorphs > ornithischians) and the geographic structure of dinosaur diversity (ornithischians > theropods > sauropodomorphs), which is also reflected in the figure order. The order should remain consistent throughout the main text unless there is a good reason for altering it.

4. In my opinion, figures are a bit bare for the reader to easily navigate through them. I suggest (a) labeling geologic stages along the "time" axis in all figures since the text refers to them frequently; (b) making the figure sections clear with "A" "B" "C" placed outside of figures; (c) adding y-axis label to all plots in Figs. 1 and 2.

5. In my opinion, the final section of the Discussion section advocating open-access veers too much away from the foci of the study. Many scientists support and respect open-access, but a substantial discussion on the matter should not inserted into the manuscript unless the study is specifically on the impact of open access. That said, it is still a fair point to bring up in the Discussion, but please keep opinionated statements to a minimum.

6. The authors should provide some insight into why diversity trends are emphasized when SQS is used on the data. Otherwise, the reader is left wondering why this could be the case.

7. Ln 118–119: "We excluded Aves as, due to the differences in the skeletons of volant taxa [...]" I don't see how this prevents one from including Aves in the study. Each dinosaur clade has distinct set of osteological attributes. Please consider removing this or elaborating on the reasoning.

Experimental design

The study addresses an important and intriguing question for paleontologists and more broadly for evolutionary biologists interested in diversity patterns through geologic time. The motivation and aims of the study are established in the manuscript. Some minor comments include:

8. I would like to see two additional sets of plots. First, diversity curves for total dinosaur samplings (i.e., Theropoda + Sauropodomorpha + Ornithischia) so that within-clade patterns can be easily compared with the pooled diversification patterns. Second, a plot of the number of new taxa vs. year. This plot will allow the reader to see patterns of new species descriptions. It would be informative to discuss the historical reasons for sudden bursts or decreases in new taxa (e.g., opening up of new fossil-bearing sites).

9. The Shareholder Quorum Subsampling should be explained briefly in the Methods section for the reader who are unfamiliar with this particular approach can get a good sense of how the values are being transformed. For example, explain how it "account[s] for differences in the shape of the taxon-abundance curve" (Ln 153–154).

10. Ln 122–125: "A potential issue with this approach [using genus-level data] is that many dinosaur genera are multispecific, but this is randomly distributed throughout our dataset and therefore should not have any substantial impact on resulting curves." Please provide a reference or supporting information to justify this statement.

Validity of the findings

11. Ln 195: "The overall shape of the raw theropod diversity curve remains consistent through publication history." I personally do not observe this in Fig. 1A, where the gap between 1991 and 2015 records fluctuate throughout geologic time. There is a large discrepancy especially in the Lower Cretaceous and latest Cretaceous.

12. I have several comments and questions regarding the correlation tests. First, what is the reason for adjusted p-values being the same across multiple publication years although the raw p-values are different? Second, were error bars for paleotemperatures incorporated into the analyses?

13. Ln 367: "In this study, we tested whether by comparing successive dinosaur diversity logistic curves we are approaching the end of the exponential phase of dinosaur diversity increase, making our diversity analyses for this clade more stable and reliable for further examination and interpretation." I may be mistaken, but I was not able to locate where the relevant results are reported in the manuscript (i.e., results from testing whether the end of exponential phase has been reached).

Additional comments

I enjoyed reading the manuscript--very interesting historical perspective on the field of dinosaur paleontology.

Annotated reviews are not available for download in order to protect the identity of reviewers who chose to remain anonymous.

---

## Round 0.2 · Major Revisions

Dear authors,

I’ve received three reviews for your ms, including two reports from reviewers who previously commented on your ms. The reviewers who previously commented on your ms have thanked the authors for their efforts to respond to their points in this revised version, and Reviewer #3 is satisfied that their main comments have been adequately addressed. Reviewer #2 has, however, expressed some concerns that although the authors have responded carefully to their points, those responses have not been translated to modifications to the text in many cases. Some of the points that remain of concern have also been raised by Reviewer #4, and I encourage the authors to please take this opportunity to fully address these constructive comments in a second set of revisions.

Reviewer 2 ·

Basic reporting

see below

Experimental design

see below

Validity of the findings

see below

Additional comments

I thank the authors for their detailed responses to my comments. I have re-read the manuscript and whilst I believe it is now slightly improved, I am a little disappointed in their defensive reaction to many of my comments, many of which remain unaddressed. I did not intend to be unpleasant, and I apologise if it came across that way. All my comments stemmed from an honest appraisal of the manuscript with a desire to help the authors improve the study. After all, I would not have bothered to write such a lengthy review if I intended to get the manuscript rejected. All replies to comments that I feel have been satisfactorily dealt with I shall not comment on here and below is a discussion of points I still feel need addressing before publication.

Comments on revised manuscript

Where you use “Judithian” you still do not explain what you mean and it is impossible to relate this to the figure unless you possess detailed knowledge of regional stratigraphy. Why not change it to Campanian-Maastrichtian instead? As it stands, many quantitative palaeobiologists (like myself) with no knowledge of local Cretaceous stratigraphy will struggle to understand what this is referring to.

The figures are slightly improved but they are still hard to follow. You need to define all the abbreviations of Epochs and Stages. Remember this is an open access journal and members of the public or non-specialists will not be familiar with the Mesozoic timescale. The grey bars for the stages are too faint. I would also prefer to have all the temporal detail on the x axis at the bottom to avoid having to jump my eyes from the numerical temporal scale on the bottom x axis to the chronostrat scale at the top x axis. You still don’t state what the relevance of the red lines are – I know they are Epoch and Period boundaries, others may not.

I am sorry that my comment about your paper not representing a filling of a “knowledge gap” caused you offense. In my defence, that is EXACTLY what the PeerJ reviewer guide asked me to assess. I formatted my review to the exact guidelines provided by the editor. As me “feeling” that the research adds little to understanding the fossil record or dinosaur evolution – that is a comment I still stand by but, as you correctly state, that is subjective, as is your opinion that it is relevant. Much of professional opinion has to be subjective, particularly in palaeontology, otherwise we would all agree all of the time in the face of irrefutable facts. Please do not claim that I am being unprofessional for disagreeing about the relevance of a piece of work. After all, I do not state that the paper should be rejected based on my subjective opinion. Nevertheless, I feel I understand the purpose of the paper to a greater degree based on your replies to many of my comments but which have not necessarily been incorporated into the revised manuscript. I would suggest that the authors carefully consider their wording when it comes to highlighting the relevance of the study and look back at some of their replies to my comments for guidance. After all, this was my main issue with the paper, I felt that you haven’t explained the relevance, you just state it.

I have not said that anything undermines the “scientific quality” of this research. I have repeatedly stated that the authors have been thorough in their analyses. My issues are with what it actually adds to our knowledge on the subject of the quality of the fossil record and of dinosaur evolution. I still think the authors needs to do a better job of explaining this in more precise terms – again, that is my main issue with this paper.

In the abstract, I still think to say that the overall shape of raw theropod diversity does not change between 1991 and 2015 is a bit of a stretch. Also, what do you mean by they are fairly evenly distributed throughout the Cretaceous? In addition, the statement saying that the “relative magnitude of these changes is greatly emphasised” after subsampling is not apparent (to me at least) in the figures. The patterns are completely different to the raw data so, how can “these” changes be greatly emphasised? I think this stems from the ambiguity of the statement. This is, in part, what I was referring to when I said the description of the results does not always match the figures.

The authors misunderstand my point about their opening line in the abstract. Of course I am not suggesting that new data doesn’t affect our knowledge! I was playing devil’s advocate by saying that the variation through historical time does not necessarily affect our understanding today, i.e. we could have had a completely different discovery sequence and still end up at the same end result at the present. I said this to try to encourage you to explain the relevance to me, rather than just stating it. In reality, it’s not really worth arguing about, but, in my opinion, it is an odd way of opening the paper. My concern here is a wording issue, nothing more.

All of the references provided by the authors for the second line of the abstract are fine and I think the revised sentence is much better. As it previously stood, the sentence suggested that conclusions had been made on raw data, which is not strictly true. That is all I was challenging and I was not attempting to take anything out of context.

In the next line, as before, I am challenging the authors to explain why it matters how the shape of diversity curves have changed through time. I am not stating that examining diversity is not important and I don’t see how the authors came to that conclusion. I concede that the authors explain this in greater depth on the introduction but it does feel like it is left hanging in the abstract. I do, however, respect that everything cannot be explained in the abstract.

In terms of using shape or trend… to me, shape also implies magnitude whereas trend does not. Therefore, switching to trend would have alleviated some of my issues over results descriptions not matching the plots (especially the theropods).

Regarding my “fool’s errand” comment. In hindsight, I see how this may have upset the authors. I apologise for this and in no way did I mean that the authors are themselves fools as, if I did, I would have to include myself in this foolish category. I have worked extensively on this question and I now believe (and that word is key), that it is not something that we can ever resolve. Of course, others are welcome to disagree on this issue.

When I state that PBDB and fossilworks are the same, I mean that they contain the same data. This was not apparent from the original sentence. I see the authors have now changed this.

I still don’t think you need the Google Scholar sentence, but that’s up to the authors. It seems a little uncomfortable at the end of the paragraph.

I still think the analysis hinted at in lines 74-77 would be more interesting than the results presented here, or even better, in addition to the results presented here.

Lines 112 onwards: you still don’t justify why you have omitted the Triassic. We have lots of Carnian and Norian dinosaurs, do we not?

Regarding the model-fitting, I still disagree. The section where you state what you are not doing starts the confusion about the model fitting. You say you aren’t doing any residual modelling then call a section model fitting where you perform correlation tests. The opening line of the model fitting paragraph also adds to this confusion where you state you perform pairwise correlation tests as your model-fitting protocol. Irrespective of the method used to detrend your data and whether this employs a model fitting approach, the main gist of this section is to test for changing correlation between diversity and extrinsic parameters. If not, it certainly reads as though it is. Therefore, the title of the section would be better changed to something like “correlation between diversity and extrinsic parameters”. For example, I have used generalised and first differencing to detrend data in some of my similar studies. I then go on to use pairwise correlation tests to assess relationships between variables. I did not call these sections of my papers “generalised differencing” or “first differencing” as these, like the ARIMA modelling here, represent the pre-processing of the data before the actual test for significance is performed. That is my issue here, you will confuse the reader by deriving your subtitle from the pre-processing rather than the actual testing.

With regard to the stratigraphic range issue in lines 134-137, I see I misunderstood. I presumed you were only counting genera that occur within a single time bin. What I presume you actually mean is, if you can’t unambiguously assign an occurrence to a single stage, then it is excluded. Is this correct? If so, I suggest you remove the reference to “stratigraphic range” which infers the range of geological time occupied by multiple occurrences of a particular lineage i.e. ranging through diversity.

In lines 143-144 you explain that you mean individual occurrences in your reply to my comment but don’t clarify this in the manuscript. As you are using the plural of occurrence it could still mean you’re deleting successive individual occurrences are whole ranges of occurrences. If you clarify to me in the reply, you should also clarify in the manuscript. This happens a number of times throughout this review. If you have to clarify to the reviewer, clarify in the revised manuscript.

I still disagree with using a parametric correlation test before/without testing for normality and don’t see the point in using more than one test in this instance. Pick the test that is best suited to the data. The violation of assumptions could lead to erroneous interpretation of results. The authors reply by saying they interpret very little from these results so it doesn’t matter. This is a worrying statement for two reasons; (i) why bother doing the test at all if you aren’t bothered about the results and; (ii) why not perform a tiny alteration to your analyses to guarantee that you are using the most appropriate test?

Lines 215-216: you’ve explained why you did this to me but, again, you haven’t altered the manuscript to explain to everyone else who will read the paper.

Line 258: I didn’t articulate my issue with this sentence very well. What I meant was how can something be “stable to changes in publication history”? – How can you change publication history? I think you mean something along the lines of “diversity is stable throughout publication history”.

Lines 280 onwards: this is a better, more cautious narrative (despite your apparent outright rebuttal in your reply to my comment). All I requested was a little more caution rather than insinuating literal biological interpretation of subsampled diversity. There is much to be said for the use of words like “apparent” when dealing with statistically manipulated data.

Regarding lines 356-357: it seems to me the authors agree with my comment attached to line 295 of the original submission – why not discuss this further as the whole nature of the paper hinges, to an extent, on the accuracy of these results, does it not? This is backed up by the fact that the authors are unwilling to plot up data where there are very few data points over fear of devaluing their manuscript. If you are interpreting these results in terms of biological interpretations (i.e. lines 361-363), you should not be silently concerned about presenting the results graphically. I understand there are already lots of plots, but they could be included as a supplement. I’m not sure what you mean about a reference in the supplement.

Regarding correlation results – I’d include a statistic for every statement. It think that’s standard practice. I don’t see the problem of including tables of stats, unless there is a strict page limit. Much better than having to chase down the supplement to check the statistics.

I don’t see how me not understanding what bibliographic publishing bias is explicitly referring to equates to another confusing statement? Perhaps I am an idiot, but it didn’t leap out to me as being obvious. The job of a reviewer is to point out ambiguities in the wording of manuscripts as much as it is to question the scientific method. Now that you’ve explained it to me I would expect you revise the sentence to explain it to the readership. Otherwise you are just dismissing my constructive comments and I might as well not review the paper at all.

In terms of the J/K boundary decline comments. I still disagree. You only see a decline across the boundary in the global data and the European theropod data. Even in the global data it is only the theropod data where it really stands out. The interval you refer to now is multiples of tens of millions of years long and I struggle to see how this can be interpreted as an “event” with any certainty at all. I don’t necessarily request you remove this all together, but don’t be misleading in the text as the figures call you out. This was the main reason I was concerned about the figures not matching the interpretation.

I still think equating food web dynamics to continental-scale diversity patterns across tens of millions of years adds very little to the paper. I’d remove it and concentrate on the message you really care about.

Lines 563 onwards – you still haven’t explained why your method is more refined than that used by Brusatte et al.? You don’t seem to explain this in the methods and I really want to know, as will many others. I’m honestly not trying to catch you out!

Lines 568-569: why not change it to the more easily understandable “more heavily sampled” then? Rather than sticking to the more ambiguous and frankly, rather odd sounding, “more-sampled”. Remember some of your readership will not have English as a first language.

Lines 686-687: This is a generalisation and over inflation of your results as you have only shown that subsampled richness is highly sensitive to changes in the taxon-abundance curve. You don’t touch on any other macroevolutionary patterns. This needs to be made clear.

I still think 686-695 is unnecessarily patronising. I think this could be condensed down to a single sentence such as “thus highlighting the need for palaeontologists to collect and publish all fossil in the field, rather than just the new specimens.”

Regarding the final paragraph, I am pleased to see that the authors have removed the final part of the sentence regarding publishing behaviour. I still think that the conclusions are somewhat underwhelming. Could you add a more definitive comment? i.e. a conclusion you could state is that we need to constantly re-evaluate the fossil record in light of new discoveries (which you do say) and, in light of this, many previous dinosaur diversity studies, are likely wrong because of the large number of new discoveries being made every year. In addition to this, the metrics presented in this paper are also likely to be shown to be wrong in a couple of years’ time. The main question remains, when will/have we settled on a biologically accurate biodiversity curve for the Dinosauria? This, I think, we will never achieve.
Concluding remarks

I do hope that this time the authors will take my comments on board as constructive criticism, rather than as harsh dismissals of their scholarship. I am happy to see a revised manuscript but, if the authors continue to dismiss my comments I see little point. To clarify, I do NOT think the manuscript deserves to be rejected as PeerJ’s mandate is to publish based on scientific merit, rather than relevance and impact. The manuscript does produce results that people in the will find useful and for that reason publication should not be delayed based on subjective disagreements between the authors and myself. This is something that should be debated openly in the scientific arena. However, it would be much better for the authors to take my comments on board and strengthen their arguments where I have challenged them before publication. Therefore, I can only recommend publication once my comments have been re-addressed.

Reviewer 3 ·

Basic reporting

The reporting has notably improved from the original submission due to the authors' revisions based on the reviewers' comments. Here are my comments for the current version:

1. First, contrary to authors stating that they have "amended the citation order throughout [the manuscript] to be in chronological order," majority of the in-text citations including multiple references are out of order. Please go through the manuscript and revise accordingly.

2. For the diversity through time plots, the gray bands indicating stages should be slightly darker to increase contrast with white bands. I was able to barely make out the bands upon printing.

3. Some claims in the Results section are not substantiated with explicit tests despite the authors using statistical terms to describe them. For example, Ln 225-226 (“the number of dinosaur occurrences published remained mostly LINEAR”), Ln 227 (“number of published occurrences has increased exponentially, and shows no sign of slowing down”), Ln 232-233 (“although the rate of growth remains approximately similar and exponential for all three groups”). These claims need to be tested statistically.

4. Discussion section: After the last paragraph, I would like to hear the authors’ thoughts on how we would know when pattern of diversity is “sufficiently” reliable?

5. Conclusions: The conclusions and major implications of this work is only broadly mentioned. Although broad statements are expected in this section, I would also like the authors to summarize and reiterate which specific aspects of dinosaur diversity patterns can we or can we not trust?

More specific and minor comments:

Ln 22, “double-dip decline”: I suggest changing the term to “two successive declines.”

Ln 113, “201-66”: Even when talking about years in the past, a range typically goes from low to high number. As such, I suggest changing it to “66-201.”

Ln 116-117, “We excluded Aves as they have a fossil record dominated by exceptional modes of preservation”: Although palaeontologists will know what you mean, the statement can be misinterpreted as the avian fossil record being replete with complete specimens. I would add a clause to clarify what you mean here (i.e., having limited occurrence of exceptionally preserved fossils will bias the results in X, Y, Z).

Ln 194, “first-order autoregressive (AR(1))”: I believe AR(1) is not brought up again in the manuscript, so I suggest removing the abbreviation.

Ln 226-227, “Since the mid-20th century, the number of published occurrences has increased exponentially, and shows no sign of slowing down”: Because it does slow down after year 2000, the sentence should be modified accordingly (e.g., “From mid- to the end of 20th century”).

Ln 381: Remove “consistent” before “patterns” because it is redundant with the word “inconsistent” earlier in the sentence.

Ln 401: Add “publication” after “0.3 and 0.5 through” for clarification.

Ln 410, “in 2015 at least a third of total dinosaur diversity”: Please substantiate this with R value.

Ln 428: Remove the phrase “in this century.”

Ln 431-433, “Further research has shown that new dinosaur discoveries strongly influence our understanding of their fossil record and diversification patterns in a phylogenetic context”: I don’t understand precisely what is meant here. Please elaborate.

Ln 434: Add “sampling” between “whether curves” to distinguish from diversity curves.

Ln 530-531, “radiation of major tetanuran and coelurosaurian clades”; Coelurosauria is within Tetanurae. Please be more taxonomically precise.

Ln 699: Change “fossils occurrences” to “fossil occurrences.”

Experimental design

Previous comments have been addressed to my satisfaction.

Validity of the findings

Previous comments have been addressed to my satisfaction.

Additional comments

Thank you for revising your manuscript appropriately to my previous comments.

Reviewer 4 ·

Basic reporting

Often vague in description of methods. Several conceptual misunderstandings. Repeatedly cites some particular literature but also misses key citations in important places.

Experimental design

Adequate in the broad choices. However, specific details are often poorly justified.

Validity of the findings

I found the Discussion to be particularly unfocussed. Many things are discussed. because of ambiguity over what is meant by 'diversity', I was not always clear about what findings the interpretations were based upon. Because of conceptual problems, and inconcsistency in author's notion of causality (e.g. the 'redundancy' hypothesis, expectations fo how the correlation of taxon counts to rock amount proxies would vary with sample completeness, see my detaled comments), I was not sure of what the broader interpretations were supposed to be.

Additional comments

I was interested to have the opportunity of reviewing this paper. The premise is a nice one. However, I agree with some of the previous referees that the paper is unfocussed. The authors did not find anything particularly useful to do with the notion of estimating changes in the pattern of fossil record diversity through historical time. So the paper comes across as a bit of a scattershot, and is not very inspiring.

Nevertheless, PeerJ is committed to publishing things that use appropriate methods and can broadly be described as science. So the paper has potential to be published here, given that more-or-less appropriate method are applied to real data. At least, that is the case in principle.

Having said that, I find the manuscript to have significant flaws relation to specific methodological choices and their explanation. There are also substantial flaws in the general reasoning and interpretation of results and their significance. There are major conceptual failings in the way that ‘diversity’ is discussed. I’ve summarised these in six points below. I have also presented a series of more detailed comments after that.

Overall, I recommend rejection with the option to resubmit. Please note that this took me a long time to get through, essentially because I had to write so much about the methods and broad-scale interpretations. So I did not write anything about the authors’ suggestions of wider implications for dinosaur diversity. This is not because I don’t have any criticisms of this Discussion sections, however.


MAIN POINTS

(1) The word ‘diversity’, as used throughout the ms, is vague and makes it difficult to follow many of the authors’ arguments. Diversity is generally taken to mean the ‘species (or genus) richness’ of an ecological assemblage, although it has another axis in the form of evenness of the abundance-frequency distribution. The authors use it interchangeably to mean two things: (i) face-value counts of dinosaur genera; and (ii) subsampled richness estimates from coverage-based subsampling (SQS). This become particularly problematic in the Discussion because the most interpretations are not clearly specified to one of the other of these time series. When it comes to interpretation of the meaning of the curves, the distinction is very important. No sane person would interpret the global, face-value fossil taxon count as a measure of diversity, but many people (in both palaeontology and in neontology) would accept the subsampled richness estimates as estimates of diversity.

(2) The conclusion regarding correlation to environmental variables is unwarranted. This seems to be based on variation in the correlation coefficient for a set of non-significant correlation tests. Non-significance indicates a non-negligible probability that the data could be generated by a null model. in other words, the correlation coefficients cannot be distinguished from zero, so a detailed discussion of variation in their strength seems ill-advised.

(3) Throughout the work, methodological decisions are incompletely or poorly justified. Some statements about methods or expectations are directly wrong, many are very unclear. I have documented most examples of this below in my detailed comments. To me, the methodological decisions do not seem to have received a due level of consideration. Some exampled that I found particularly difficult to swallow: use of time bins that have uneven duration, comparison of time series using ordinary correlation methods rather than time series methods or first-differenced data, analysis of genus-level data. I believe that these decisions need to be revisited, and the analyses should be conducted again.

(4) In several places, expectations are stated that I do not find convincing. For example, they seem to equate stability of diversity curves through historical time with ‘redundancy’. however, this could in fact indicate that subsampling methods work well. They also state that as sampling improves that dinosaur diversity will deviate further from patterns of rock outcrop area, but I would predict the opposite (as we sample the rocks more completely, the geographic/temporal record will more closely resemble the availability of rocks). Finally, they suggest that a non-random correlation with sea level (or other environmental variables) could be present early on in sampling (when counting error is expected to be high), but then disappear as sampling continues. In fact, unless there is some reason for counting error to be systematically associated with sea level, rather than random, these is no basis on which to suggest this.

(5) Finally, in analysis of the same dataset, Benson et al (2016) found that something like 80% of variation in apparent ‘global’ subsampled species richness estimates could be exampled by variation in the palaeogeographic spread of fossil localities. In other words, ‘global’ palaeodiversity just tracks expansions and contractions on the geographic area that is available to be studied (in this case, the addition/subtraction of continents). This casts serious doubt on the value of ‘global’ diversity estimates from this particular database. In fact, only the regional studies are appropriate. since this is done and dusted in the previous literature, I recommend that the authors ditch their ‘global’ analyses and present only the regional results.

(6) In the Introduction, ‘non-random’ addition of taxa to the published occurrence pool is discussed. But no mechanisms are given for this. Later on, in the Discussion, the authors discuss publication bias on new taxa. But this needs to be included when the initial hypotheses are set up too.



DETAILED COMMENTS

ABSTRACT

“…account for ecological changes in the shape of the taxonomic abundance curve.”
>Delete ‘ecological’.

“In all three groups, the shape of raw global diversity through publication time remains intriguingly consistent”
>Delete ‘intriguingly’. It is uninformative.


“The continental signal reflects this global pattern too”
>Requires clarification. ‘continental’ could be taken to mean ‘on land’ or ‘not on islands’ but I believe the authors mean the regional patterns on individual continents.

“Our results suggest that historical changes in database compilation, particularly in terms of the publication of additional specimens of previously identified species, affects the relative magnitude of macroevolutionary patterns for dinosaurs and our interpretations of the processes that govern them.”

>This is unsatisfying to me. It doesn’t seem to say anything other than as publications accrue, our knowledge of diversity patterns changes. This is self-evident. Is there some wider message the authors wish to convey such as whether the signal improves? Or whether bias is reduced (or increased…)? More specific conclusions would help to give the work wider relevance.


INTRODUCTION

“…palaeobiology underwent a renaissance…”
>Should cite work by others such as Gould, Stanley, Valentine, Van Valen. Currently only cites a narrow range of topics published on by Raup and by Sepkoski.

“one of the most crucial aspects of palaeobiology”.
>Should state what this is crucial to. For example, what does it allow palaeontologists to do? Or what wider questions of broad scientific interest does it help to address. ‘Crucial’ on it’s own is uninformative.


“All of these studies, both older and more recent, are under-pinned by a single principle, in that they rely on counts of the number of taxa present through geological time.”
>This is not really true. They rely on recorded occurrences of fossil taxa in one form or other. one of the things allowed by this is counting. But many studies don’t involve counting taxa in the strict sense.


“broadly termed as ‘bias’. This includes factors such as sampling intensity, different sampling availability, and variable depth of taxonomic research (Benton 2008a; Benton 2008b; Raup 1972; Raup 1976; Tarver et al. 2011)”
>I don’t feel happy that this statement cites work by Mike Benton. Benton seems broadly to support the idea that biases are not important. How about instead citing work my Andrew Smith, or Uhen & Pyenson?

“This analysis was based on comparing two records”
>Rephrase. This could be misread as entailing two *occurrences* rather than two compendia.

“Alroy… further showed that database age does have an influence on North American mammal diversity estimates, but this apparent phenomenon has been largely ignored since then”.
>Should add more relevant literature. Alroy (2010) showed that the PBDB generated similar genus counts to Sepkoski’s compendium. Sepkoski et al’s ‘consensus paper’ showed that many different compendia of diversity gave similar curves too. Benton’s stuff about dinosaur collector curves might be relevant too.

“because independent datasets yield similar diversity curves, this suggests that convergence on a common signal indicates accuracy”
>Either that, or it indicates that all databases derive from the same set of real-life fossil occurrences. Arguably, this says nothing about accuracy, a point that has been made on several occasions in review papers by Andrew Smith.

“Many of these studies employ subsampling methods that are sensitive to changes in the shape of the taxonomic abundance distribution, which we would expect to change in a non-random fashion based on new discoveries through time as they are published (Benton 2015; Benton et al. 2011; Benton et al. 2013).”
>I think this requires more explanation. I agree that these should change ‘non-randomly’. This principle is well established in fact (see many papers on ecological census data, and both Alroy’s and Chao’s implementations of equal-coverage subsampling (SQS). In particular, all things being equal we’d expect the proportion of singletons to decline as occurrences accrue. This is supposed to happen and would not introduce an problem - but the text here implies (indirectly) that these would be a problem, and cites several firebrand papers that take a hatchet to any statistical sense or reason. if you think there is some other way in which publication causes occurrences to accrue that would actually bias the record, please explain that here. The current text is more or less uninformative as to what the problem could be.

“we expect the shape of raw diversity curves to change (either uniformly, randomly, or structurally) through time in concert with new taxonomic discoveries and as sampling increases (Alroy 2000b; Sepkoski 1993)
>I don’t know what is meant by ‘structurally’, so I can’t comment on that. But I wouldn’t expect the shape to change ‘randomly’, minimally it would be a time series who’s current state is influenced by its past state. I guess it could change according to a random walk, but it isn’t clear to me what this would mean in the context of abundance-frequency distributions. Similar comments apply to ‘uniformly’ (should the abundance-frequency distribution be uniform, or change uniformly? If it changes uniformly, what does this mean?). I’d like the authors to think more about this sentence and rewrite it to better express what they mean.

>I don’t agree with the sentences following this one. If the methods work, then we expect them to produce estimates of species richness when the data are informative. When the data are uninformative, not estimates are produced, or estimates have a high error associated with them. If adding occurrences adds accuracy to the estimates then we’d expect the subsampled diversity curve to ‘settle’ at some point in historical time. This is assuming that the methods work as they should. So far in the ms, the authors have not proposed any reasons why the methods might not work, so we have no reason to suspect that the diversity curve would change substantially through publication time. I really think this is an important missing piece of explanation.

>The authors suggest that correlations with variables like sea level might disappear as occurrences accrue through historical time. But they propose no concrete reason to expect this. All things being equal, estimation error will be highest early in the history of study. Unless the error is non-random, and specifically non-randomly associated to sea level change, then the probability of this causing a spurious correlation with sea level is low. Error generally erodes non-random signal rather than introducing it - except in the case of systematic bias. If the authors think that a systematic bias is at work then they should explain that.

>Furthermore, this section is clearly setting the authors up for their later discussion of correlations with sea level. However, no significant correlations with sea level are found in this paper, so it is immaterial whether correlations should be expected to ‘appear’ or ‘disappear’.


“results from analyses based on model-fitting or pairwise correlation analyses with extrinsic factors such as sea-level or palaeotemperature (Butler et al. 2011; Mannion et al. 2015; Martin et al. 2014; Nicholson et al. 2015; Tennant et al. 2016a; Tennant et al. 2016b) will change.
>This statement mis-cites several papers as demonstrating links between diversity and environment, when they in fact provide no such evidence. Butler et al (2011) rejected correlations with environmental variables and should not be cited as finding evidence of them here. Similarly, Nicholson et al (2015) found no such correlation. Rather, they proposed some regionally heterogeneous link to environment based on qualitative appraisal. The ‘correlations’ found by Tennant et al. are non-significant, and would cause any reasonable person to state that there was no strong evidence of correlations of diversity with sea level.

>How about citing Some of Shannan Peter’s stuff about common cause in marine invertebrates, of Mayhew et al on correlations of subsampled marine invertebrate diversity with temperature? Also relevant is work showing links between marine tetrapod diversity and sea level. There is other work out there too.

“ the correlations reported by the first study, also non-replicable by (Mannion et al. 2015) and (Tennant et al. 2016a), were fairly unstable even based on very recent changes in taxonomy”
>This doesn’t provide a good example. In fact, Martin et al (2014) did not even find a correlation of croc diversity to temperature. Have a look, it’s true.
>I’m concerned also when easing this section, and the Discussion. The authors state that Jouve et al showed that very small changes in taxonomy could change the apparent correlation with environmental variables. These is a missing piece of information which should say: ‘very small changes in a very small dataset of Late Jurassic/earliest Cretaceous thalattosuchians (members of one small clade of crocs). It is necessary to make this clear to avoid apparently undermining the entire endeavour of richness estimation and comparison to earth system variables.

“In this study we do not discuss the following: (1) what time-binning methods are appropriate for the fossil record; (2) what analytical methods are optimal for accounting for the incompleteness of the fossil record, and (3) what the impact of temporal variation in the rock record through space and time have on our understanding of diversity, and these factors are appropriately discussed in more detail elsewhere”

>Some of this is OK. But other aspects are not. For example, there is no ‘global’ fossil record, and no justification for analysing it as such. The real fossil record comprises just a few well-sampled, informative regions in space and time. These are separated by black nothingness, as is evident on a palaeomap such as generated online by the PBDB. The authors appear to show little consideration of some important issues that could influence their results. find this hard to justify.

“We excluded Aves as they have a fossil record dominated by exceptional modes of preservation (Brocklehurst et al. 2012; Dean et al. 2016)”.
>Does Dean et al (2016) discuss the avian fossil record. This seems to be a paper on pterosaurs. The author of the current paper seem to be following the methodological decisions of Benson et al (2016), but do not cite that work.

“Despite the fact that some dinosaur genera are multispecific, it has been shown previously that both genus- and species-level dinosaur diversity curves are very similar (Barrett et al. 2009)”.
>Also here. Benson et al (2016) analysed the same dataset as analysed here, and did so at both genus and species level. They showed that some elements of the pattern recovered are the same at different taxonomic levels (slope of regression against time). But when you actually compare the genus- and species level dinosaur curves they actually seem quite different. So is this statement justified? I don’t think so.

“there is more error in species level dinosaur taxonomy than for genera (Benton 2008b)”.

>I don’t think benton really provided strong evidence of this in the first place. But let’s argue that he did. Even if that were true, a lot of taxonomic work has been done since 2008, and almost all dinosaur genera are monospecific, so this it is hard to believe that dinosaur species are inadequately defined. I actually think many ‘wastebasket’ (undiagnostic) occurrences have historically been assigned to genera based on no evidence, and are represented as occurrences of genera in the database. You can check out taxa like Megalosaurus and Tyrannosaurus if you want to see evidence of this. So i don’t think there is a strong argument that the species-level data are worse than the genus-level data.

“For our study, it is less important what time binning scheme we use relative to the consistent treatment of it across different publication intervals”
>To say it is ‘less important’ is clearly not enough. In fact, to justify this you’d have to show that using intervals of highly unequal lengths causes no additional harm. Clearly it could cause harm: the strength of correlations between the diversity curves from data of different historical times should be inflated if no attempt is made to standardise interval durations.


METHODS

“defined as 1 minus the number of singleton occurrences over the total number of occurrences (Good 1953)”
>This could be read as (1-singletons)/(total occurrences). Clarify explanation.

“A coverage value of zero indicates that either all taxa are singleton occurrences, or that there are simply no taxa to sample”
>It baffles me why one would estimate coverage for a taxon pool that includes zero taxa. Suggest deleting: “…or that there are simply no taxa…”.

“The advantage of SQS here, then, is that it returns estimates of diversity even at very low sampling levels”.
>This is not correct unless you set the quorum very low, which is inadvisable as the estimates would be highly inaccurate. This would be particularly problematic as most authors using this method do not present error bars on their richness estimates.

“For each time bin, u is divided into the quorum level (Alroy 2010a), thereby providing an estimate of standardised diversity in a manner that is flexible in response to changes in the shape of the taxon occurrence distribution”
>This is incorrect, the division by u does not results in ‘flexible in response to changes in the shape of the taxon occurrence distribution’. I’m not even sure what is meant by this. In fact, it is a correction factor intended to transform coverage of the sampled occurrence pool into coverage of the underlying, total, occurrence pool.

“In all subsampling replicates, singletons were excluded to calculate diversity (but included to calculate Good’s u), as they tell us little about the underlying taxon distribution (i.e., a linear relationship of 1:1), and can distort estimates of diversity”
>I don’t follow the logic here. Part of the problem is that I don’t know what is meant when the authors say ‘a linear relationship of 1:1’. What variables is the relation describing? Has this practise been used in any previous work that should be cited?

“Dominant taxa (those with the highest frequency of occurrences per bin) were included, and where these taxa are drawn, instead of their share contributing towards the quorum, 1 is added to the subsampled diversity estimate for that bin (Alroy 2010c)”
>The word ‘instead’ seems out of place here. Each taxon that is drawn during subsampling ads 1 to the richness estimate, including, but not limited to, the dominant (most abundant) taxon. This sentence could be read as saying that an additional 1 (i.e. 2) is counted for the most abundant taxon.

“1000 subsampling trials were run for each dataset (Theropoda, Ornithischia, and Sauropodomorpha), and the mean diversity reported for each publication time interval”
>Please check that the mean, and not the median, is indeed being used.

“We set a baseline quorum of 0.4, as this has been demonstrated to be sufficient in accurately assessing changes in diversity (Alroy 2010a; Alroy 2010c; Mannion et al. 2015; Nicholson et al. 2015; Tennant et al. 2016a).”
>It actually hasn’t been shown that this results in ‘accurate’ richness estimates, although it has been shown that relative diversities obtained at a quorum value of 0.4 are similar to those at higher quora. To my knowledge, only Alroy (2010; Palaeontology; Science) shows this, not the oner cited papers.

“This dual method is important, as not all publications name new taxa – some add to our knowledge of existing taxa by publishing on new occurrences or collections, and therefore by applying a method that accounts for changes in taxonomic abundance we can see how publication history influences diversity through subsampling methods”.
>I’m not confident that the authors have understood the method/database well. Each occurrence is only recorded to one publication in the dataset (this is evident from looking at rows of the data downloads from Fossilworks and the PBDB, so a second publication on the same occurrences of Allosaurus in site X would not qualify Allosaurus to become a non-singleton (contrary to the author’s statement above). The ‘publications’ method of assessing singletons requires *both* that at least two localities include an occurrence of taxon Y *and* that at least two of those occurrences are attributed to different publications.

“For our model-fitting protocol, we follow the standard procedure outlined in numerous recent analytical studies, by employing simple pairwise correlation tests to the residuals of detrended time series at the stage level“
>This sentence states that detrended data series were analysed. But the following sentence suggests otherwise (GLS was used to to model the serial correlation).
>Also note that use of ARMA models on fossil time series data originates with Hunt et al (2005; species-energy relationship) so far as I know. It was also used by Marx & Uhen (cetaceans) before the cited works.

“This method eliminates the potential influence of any long-term background trend”
>This is actually not true of AR1 models. AR1 models *only* model the serial correlation term.

>The text suggests that the authors fit an AR1 model, and only an AR1 model. However, the serial correlation structure of the data is not known in advance. So they should be using information criteria (AICc) to compare AR1 models to OLS (=AR0), AR2, and possibly other time series models. Otherwise, there is no basis on which to assume that AR1 is the best model of the relationship between variables.

“for palaeotemperature we used the data from Prokoph et al. (2008)”.
>Which data? Prokoph provided data for several stable isotopes in several latitudinal classes (low-, mid-, and high-latitude).

“We performed pairwise correlation tests between our diversity estimates and each environmental parameter using parametric (Pearson’s product moment correlation coefficient [r]) and non-parametric (Spearman’s rank [ρ]) tests.”
>This is inappropriate given that the data are time series. The authors should be using time series methods (e.g. ARMA models) as they used above. Either that or using first-differencing possibly. Using ordinary correlation metrics on untransformed data is inappropriate for the reasons explained in the preceding section by the authors themselves.

“Differently to Tennant et al. (2016b), we excluded the first 5 Jurassic data points from our analyses instead of treating them as missing data”.
>No reason is given why any special treatment needs to be made of Jurassic data. Please explain what is wrong with the data. Also, I’m not certain of the difference between ‘excluded’ and ‘treated as missing data’. So far as I know for all the analyses discussed so far the practical implementation would be identical.

“All analyses were carried out in R version 3.0.2 (R Development Core Team 2013).”
>Cite any packages used here.


RESULTS

Throughout the results, the authors should make a distinction between ‘diversity’, and ‘face-value taxon counts’ (or similar). This is important. ‘Diversity’ is a parameter of real ecological assemblages. Counts of taxa may or may not be similar to diversity. However, my sense is that most people would not view counts of fossil taxa this way. Subsampling methods attempt to estimate relative diversities of occurrence pools and could reasonably be referred to as ‘diversity’. But they would be better as ‘subsampled diversity estimates’.

Allegedly ‘global’ analyses are nonsensical as their apparent changes in diversity can result just from addition/subtraction of the number of continental regions sampled. This has been shown compellingly by Benson et al (2016; cited in text) and Close et al (2017; Nature Communications) for the same exact data as analysed by the authors of the present work. There is no useful reason to present ‘global’ diversity curves.


“The reason for this distinction between SQS and raw diversity is that subsampling is sensitive to changes in the species abundance pool, and thereby reduces the impact of intensely sampled time intervals such as the latest Cretaceous”
>This is the second time that the authors have described SQS as ‘sensitive to changes in the species abundance pool’. While this is broadly true, the statement does little to explain what the method is trying to do. I’d advise writing something more informative such as ‘this subsampling method estimates diversity by standardising coverage of the occurrence-frequency distribution…’

“The Barremian to Cenomanian is approximately constant in diversity, and consistently increasing through time to become more decoupled in magnitude”
>I do not know what is meant by ‘decoupled in magnitude’. Please clarify in text.

Lines 390-411: Twenty lines of text are expended discussing the size of a correlation coefficient for a non-significant correlation. The correct interpretation of non-significance is that the correlation coefficient cannot be distinguished from zero. So this text is unwarranted and misleading. Furthermore, no table is cited in-text so it is difficult to consult the correlation results. Some aspects are not clear in the text. For example, is the correlation with total dinosaur diversity significant or non-significant?


“As research on dinosaurs continues in this century and new taxa are described from existing fossiliferous formations, one implication of this is that raw diversity is expected to become less correlated with rock availability as result of increasing sampling effort (Benton 2015; Raup 1977; Wang & Dodson 2006)”
>This is clearly not the expectation. The total rock outcrop provides us with the maximal dinosaur fossil record. Once we have sampled all the available rocks completely then the record should exactly effect the availability of rocks (and proxies for that, such as outcrop area). I’m saying we should expect that raw diversity should become more strongly correlated with outcrop area as those outcrops become more completely sampled.


“The fact that the curves remain relatively linearly consistent despite the non-random addition of new taxa also provides support for the ‘redundancy’ hypothesis, that fossils and sampling are non-independent from each other, when only raw data are considered (Benton 2015; Benton et al. 2011; Benton et al. 2013; Dunhill et al. 2014).”

>It is clear that sampling leads to fossil discovery, causality cannot flow int eh opposite direction to this. This premise is the basis of the subsampling methods used by the authors in the present work, and across a vast ecological and paleontological literature. The ‘redundancy’ hypothesis in its most general form is therefore clearly spurious at an axiomatic level. That is not to say that it does no apply in some narrow cases (Benton’s classic example was counting anomodont-bearing formations). Some authors have stated that high fossil richness leads to high levels of sampling, and framed this in the context of redundancy. However, this case is also subtly different - the abundance of fossils and the species richness of the underlying occurrence pool are different to each other. Abundant fossils could represent occurrence pools with either high or low richness.


“While others are now reaching the same conclusion, at least for the Mesozoic tetrapod record, this further suggests that ‘correcting’ diversity estimates by using static proxies for sampling is not an appropriate methodology (Benton 2015; Brocklehurst 2015; Sakamoto et al. 2017).”
>This statement is very sweeping. The cited works have criticised the model-based methods for several distinct reasons. They do not all use the argument of redundancy, which is implied by the statement ‘while others are now reaching the same conclusion’. Furthermore, this sentence seems out of place. The model-based methods have not been introduced or explained in the text so this sentence dis somewhat orphaned. I don’t believe that an uninformed reader would know what it is supposed to refer to
>Finally, and most impotently: I don’t see that this is a logical consequence of the results found by the present authors. Stablity of subsampled diversity curves could just indicate that the subsampling methods work well. Stablity of the face-value counts could indicate that people continue to apply the same collection effort to outcrops of the same ages through historical time.


“This raises questions about the extent to which many aspects of diversity curves could be artefacts caused by changes in global sea levels, tectonics, and other geological processes related to preservational or geological megabiases (Heim & Peters 2011; Peters & Foote 2001; Peters & Heim 2010; Peters & Heim 2011; Smith & Benson 2013; Smith et al. 2001; Smith et al. 2012; Smith & McGowan 2007).”

>I agree that this question is important. It is logical to suggest that rock availability causes face-value taxon counts, and could cause subsampled ‘global’ richness estimates. However, by strongly advocating the redundancy hypothesis earlier in the text, the authors seem already to have trashed this hypothesis. There is a schizophrenic nature to the discussion because earlier they seem to agree with a body of literature that attacks correction of taxon counts by modelling of rock amount proxies based on a reversed causality argument (redundancy).

“We find that there are three main time periods when great caution should be applied to interpreting further processes or patterns based on dinosaur diversity, especially at a global level.”

>Here, and elsewhere, it is not clear whether the authors are talking about the face-value taxon counts or the subsampled diversity estimates. Clearly, great caution should be applied to face-value taxon counts. In fact, I’d say that you can’t use face-value taxon counts as diversity estimates. This should be obvious on common-sense grounds. The authors need to clarify what is mean by ‘diversity’ throughout the Discussion, and the rest of the manuscript.

>I’d advocate taking out the ‘global’ curves for reasons stated above. I don’t think ‘caution’ in their interpretation is required in a handful of intervals. I think it is required in all intervals.

---

## Round 0.3 · Minor Revisions

Dear Dr. Tennant,

Further to the extensive revisions required for the last round of review, and given that you introduced comprehensive changes and an additional co-author, I felt it necessary to send your manuscript out again for review. Unfortunately I was not able to receive comments back from the previous reviewers, and therefore I had to request a new report. I understand receiving comments from a new reviewer at this late stage can be frustrating, and I share the authors frustration, however my priority is to ensure the paper is sound. Following my own review of the manuscript, I felt it was necessary to receive a second opinion from a reviewer more familiar with the dinosaur record. I hope the authors understand my position, and I request that they please consider the additional, constructive changes that the reviewer has suggested.

It is my impression that the authors have devoted considerable effort to all the previous revisions requested and that the manuscript is much improved, and will be suitable for publication in PeerJ.

·

Basic reporting

In general, I followed the course of the paper. The structure and referencing are sound, the text mostly unambiguous, and the relevant raw data available. I did, however, note a few typos, omissions and ambiguities. These are addressed, by order of their appearance in the text, below:

Line 64-65: “(e.g. (Sepkoski et al. 1981); Sepkoski Jr (1993); Alroy (2000b))” – the bracketing around the first reference here is inconsistent with the second two.

Line 69-70: “At the present, the first argument appears to be the best supported by analytical evidence” – I feel that this needs its own references explicitly tied to it, as it is something of a bold statement.

Line 131: “(November 2017; note a new download was performed subsequent to peer review)” – When was the new download performed? Or was that the download performed following peer review? If the former is the case, were all the analyses repeated on the new download? If so, why report the previous one?

Line 169: “We stopped at 1991” – Is there any particular reason for this cutoff? Was it due to computational constraints, deemed a sufficient sample, or is something about the post-1991 record considered fundamentally different to the dinosaur record from before then? I think all of these positions are defensible, but would like to know the author’s thought process on the matter.

Line 413: Typo: should read "steadily increasing" or "steady increases in".

Line 414: Typo: should read Jurassic here instead of Cretaceous (I think?).

Line 435: “meaning that we cannot interpret anything from these results with any high level of confidence” [emphasis added] – Really, this means that we cannot interpret anything with any level of confidence – it is 95% or nothing, after all.

Line 489: It should probably be noted in the discussion of it here that the Smith & McGowan (2007) method has seen substantial criticism (e.g. Sakamoto et al., 2016) - it would help inform the next few lines discussing about how this influence has apparently been mitigated since.

Line 554: There is a break between two sets of references here – I think that it is just a single series of citations that has been broken, but it may be that a sentence has been deleted. Please check.

Line 627: “This find is somewhat contrary to that of Sakamoto et al. (2016)” – Do you have any idea as to where this discrepancy may come from? I suppose it may be a result of the temporally biased nature of sauropod phylogenies – relatively few titanosaur taxa were included in phylogenies as of 2016, and so may result in underestimated speciation/cladogenesis rates. It may be relevant to consider imperfect phylogenetic coverage as a point in support of count-based diversity metrics, even when they require correction for sampling. Alternatively, it may represent a taxonomic decoupling between genera and per-species dynamics. I appreciate this is all currently speculative, I was just curious.

Figures
The time (x) axis in Figures 4-12 is quite cluttered. Maybe it would be clearer if they were modified to feature a standard, coloured, geological timescale as the x-axis?

Experimental design

The research question that the analyses were conducted to address is well-identified. The methodology is generally transparent, save for one ambiguity I would like to clear up.

Lines 149-150: “These databases are based on a comprehensive data compilation effort from multiple workers and represent updated information on dinosaur taxonomy and palaeontology at this time.” – Just to be clear, how do your (e.g.) cumulative frequency plots of named genera (Figure 2) deal with taxa currently considered invalid? Are only taxa currently considered valid included at all; does the curve simply reflect the total number of named genera regardless of current validity; or are all taxa named within each time bin included, but with those later deemed invalid subtracted from the appropriate time bins? Taxonomic revision represents an (arguably) equally important contributor to our developing understanding of diversity patterns as does discovery, so this is highly significant and so must be stated as clearly as possible.

Currently, my understanding is that for the occurrence-based analyses you are using the taxonomy as of 2015 in all of the two-year bins? This would be appropriate as it eliminates another potential variable to focus on sampling, but this needs to be explicitly stated in the main text to ensure that the reader does not confuse it with a comparison of the ‘state of knowledge’ of dinosaur diversity through publication history. An additional cumulative plot of genera considered valid in each bin may add some nuance to Figure 2, though.

My more substantial concern with the experimental procedure conducted herein is that bootstrapping of results, in order to calculate confidence intervals, was apparently not performed. This is expanded upon in section 3 below.

Validity of the findings

I am satisfied that the results presented herein are broadly valid and agree that they are worthy of publication. I applaud the authors for publishing negative results, something universally acknowledged as important but relatively rarely actually performed. I also think that this study will be useful as a starting board from which further analysis can look into other potential biases, as discussed in the text. Most importantly, I think that the chief conclusion – that ‘global’ signals are highly uncertain due to their nature as a gestalt of heterogeneously-sampled and unstable regional data – is both valid and significant to a field in which ‘global’ curves remain commonplace.

However, I am concerned in that only mean subsampled diversity values are reported and compared, with 95% confidence intervals apparently not calculated. This is highly problematic when the results are then used to lend support to purported evolutionary events across the Jurassic-Cretaceous boundary (e.g. 519-521): even error bars on the ‘global’ results would help to allow this claim to be appropriately evaluated.

I appreciate that it is already obvious that there will be low statistical power in many time bins in particular regions, and that estimating error is less likely to have an impact on interpreting the main results comparing subsampled diversity between 1991 and 2015, given that results are generally similar aside from very different individual points. Wide error bars are already pre-empted by the chief conclusion that ‘global’ patterns are highly uncertain due to instability in individual regions but including them would help quantify this. In addition, given that an implicit practical aim of this study is (presumably) to indicate where in the sampling curve is “good enough” to estimate diversity in an exemplar case, would that not most appropriately be handled by inspecting when the 95% confidence intervals first show substantial/total overlap with those calculated for data from 2015?

Additional comments

In summary, although I do see virtue in the approach and interest in the results that would merit publication, I do still have some concerns that I, ideally, would like incorporated into the analyses, and at the very least acknowledged in the discussion. My primary concern is the lack of consideration of confidence intervals in the interpretation of a genuine biological signal across the Jurassic-Cretaceous boundary: I also would appreciate clarification on how changes in taxonomy were handled.
Still, I do think that the results herein are worthy of publication. Although there is partial overlap with some previous studies, collaboration of previous works and dissemination of negative results are both uncommon yet important in science. In particular, the main conclusion from this study (that “global” diversity patterns in the fossil record are highly problematic) is one that really cannot be stated enough. I hence feel that I can recommend it for publication in PeerJ, provided that the above comments are satisfactorily addressed. Still, as these revisions will require re-analysis in order to generate confidence intervals, I would appreciate the opportunity to see it again.

---

## Round 0.4 · accepted · Accept

Dear Dr. Tennant,

Thank you for addressing the comments raised by the referee and the editorial suggestions to the text in the previous round of review. I look forward to seeing your article published.